# Evaluating CHASERV4.0 global formaldehyde (HCHO) simulations using satellite, aircraft, and ground-based remote sensing observations

Hossain Mohammed Syedul Hoque[1], Kengo Sudo[1,2], Hitoshi Irie[3], Yanfeng He[1], and Md Firoz Khan[4]

[1]Graduate School of Environmental Studies, Nagoya University, Nagoya, 4640064, Japan

[2]Japan Agency for Marine-Earth Science and Technology (JAMSTEC), Kanagawa, 2370061, Japan

[3]Center for Environmental Remote Sensing (CEReS), Chiba University, Chiba,2638522, Japan

[4]Department of Environmental Science and Management, North South University, Bangladesh

*Correspondence to*: Hossain Mohammed Syedul Hoque (hoquesyedul@gmail.com; hoque.hossain.mohammed.syedul.u6@f.mail.nagoya-u.ac.jp)

**Abstract**

Formaldehyde (HCHO), a precursor to tropospheric ozone, is an important tracer of volatile organic compounds (VOCs) in the atmosphere. Two years (2019 -2020) of HCHO simulations obtained from the global chemistry transport model CHASER at a horizontal resolution of $2.8° \times 2.8°$ have been evaluated using the Tropospheric Ozone Monitoring Experiment (TROPOMI) and multi-axis differential optical absorption spectroscopy (MAX-DOAS) observations. In-situ measurements from the Atmospheric Tomography Mission (ATom) in 2018 were used to evaluate the HCHO simulations for 2018. CHASER reproduced the TROPOMI-observed global HCHO spatial distribution with a spatial correlation (*r*) of 0.93 and a negative bias of 7%. The model showed good capability for reproducing the observed magnitude of the HCHO seasonality in different regions, including the background conditions. The discrepancies between the model and satellite in the Asian regions were related mainly to the underestimated and missing anthropogenic emission inventories. The maximum difference between two HCHO simulations based on two different nitrogen oxide ($NO_x$) emission inventories was 20%. TROPOMI's finer spatial resolution than that of the Ozone Monitoring Experiment (OMI) sensor reduced the global model–satellite root-mean-square-error (RMSE) by 20%. The OMI and TROPOMI observed seasonal variations in HCHO abundances were consistent. The simulated seasonality showed better agreement with TROPOMI in most regions. The simulated HCHO and isoprene profiles correlated strongly ($R = 0.81$) with the ATom observations. However, CHASER overestimated HCHO mixing ratios over dense vegetation areas in South America and the remote Pacific (background condition) regions, mainly within the planetary boundary layer (<2 km). The simulated seasonal variations in the HCHO columns showed good agreement ($R > 0.70$) with the MAX-DOAS observations and agreed within the 1-

sigma standard deviation of the observed values. However, the temporal correlation ($R$~0.40) was moderate on the daily scale.CHASER underestimated the HCHO levels at all sites, and the peak occurrence in the observed and simulated HCHO seasonality differed. The coarse model resolution can potentially lead to such discrepancies. Sensitivity studies showed that anthropogenic emissions were the highest contributor (up to ~35%) to the winter-time regional HCHO levels.

# 1 Introduction

Formaldehyde (HCHO), the most abundant carbonyl compound in the atmosphere, is a high-yield oxidation product of all primary biogenic and anthropogenic non-methane volatile organic compounds (NMVOCs). Methane ($CH_4$) oxidization produces background HCHO concentrations of 0.2–1.0 ppbv (Burkert et al., 2001; Singh et al., 2004; Sinreich et al., 2005; Weller et al., 2000). Along with secondary sources (i.e., oxidization of NMVOCs), biomass burning, industrial processes, and fossil fuel combustions are primary HCHO emission sources (Fu et al., 2008; Hak et al., 2005; Lee et al., 1997). However, the oxidization of NMVOCs drives the spatial variability of HCHO on a global scale (Franco et al., 2015). The HCHO removal mechanisms include photolysis at wavelengths below 400 nm, oxidization by hydroxyl radicals (OH), and wet deposition. The atmospheric lifetime of HCHO is around a few hours (Arlander et al. 1995). Therefore, HCHO observations can help elucidate chemical processes in the atmosphere. A few examples are the following: (1) the ozone ($O_3$) production regime can be determined from the HCHO to nitrogen dioxide ($NO_2$) ratio (Duncan et al., 2010; Hoque et al., 2022; Martin et al., 2004); (2) midday OH levels can be quantified from the oxidation of isoprene into HCHO (Kaisar et al., 2015); and (3) HCHO, being an intermediate product in oxidation chain of NMVOCs, engenders the formation of carbon monoxide (CO) and carbon dioxide ($CO_2$). Consequently, CO chemical production from NMVOCs and $CH_4$ can be quantified from HCHO measurements (De Smedt et al., 2021).

Given its importance, global HCHO observations started in 1995 with the launch of the nadir viewing ultraviolet (UV) sensor Global Ozone Monitoring Experiment (GOME; Burrows et al., 1997). Since then, numerous sensors have succeeded:  SCanning Imaging Absorption Spectrometer for Atmospheric CHartographY (SCIAMACHY; De Smedt et al., 2008, 2010; Wittrock et al., 2006) onboard the ENVISAT satellite, Ozone Monitoring Instrument (OMI) (Levelt et al., 2018), Global Ozone Monitoring Experiment – 2 (GOME-2) (Munro et al., 2016), and Ozone Mapping and Profiler Suite (González Abad

et al., 2016, new reference ). The HCHO observations from these sensors have been used extensively to evaluate models, air quality, and climate change (Chutia et al., 2019; De Smedt et al., 2010, 2012, 2015; Hoque et al., 2022). The Tropospheric Ozone Monitoring Instrument (TROPOMI) (De Smedt et al., 2021; Veefkind et al., 2012), launched on the European Copernicus Sentinel-5 Precursor (S5P) satellite on October 13, 2017, is the recent addition to the series of nadir viewing UV sensors providing HCHO data. The unprecedented original spatial resolution of $3.5 \times 7$ km$^2$ (across-track $\times$ along-track) refined to $3.5 \times 5.5$ km$^2$ on August 6, 2019, is the crucial feature of TROPOMI. Such spatial resolution is almost 16 times finer than its predecessor, OMI (De Smedt et al., 2021). Such high-resolution observations will likely reduce uncertainties in the HCHO products for multiple research purposes.

Several studies using the TROPOMI HCHO product have been reported in the literature. De Smedt et al. (2021) and Vigouroux et al. (2020) have validated TROPOMI HCHO comprehensively against MAX-DOAS and FTIR networks. Both studies have concluded that TROPOMI HCHO products have achieved the pre-launch accuracy requirement of $< 40$–$80\%$. Ryan et al. (2021) and Chan et al. (2020) reported good agreement (temporal correlation, $R > 0.70$) between TROPOMI and MAX-DOAS in Melbourne and Munich. In addition to validation studies, HCHO products have been used to infer changes in the global HCHO levels during the COVID-19 pandemic-led shutdown (Level et al., 2022; Souri et al., 2021; Su et al., 2021), demonstrating the role of anthropogenic emission on global HCHO variability.

Among the multitude of applications of TROPOMI HCHO observations, few efforts have specifically evaluated HCHO simulations from global chemistry transport models (CTMs). This work evaluates the global Chemical Atmospheric General Circulation Model for the Study of Atmospheric Environment and Radiative Forcing (CTM CHASER) (Sekiya & Sudo, 2014; Sudo et al., 2002, 2007) simulated HCHO spatiotemporal distribution against TROPOMI HCHO observations. In addition, airborne and ground-based observations are used to validate the simulated HCHO profiles and surface mixing ratios in a few regions. CHASER simulations of NO$_2$, OH, and O$_3$ have been evaluated against satellite and ground-based observations (e.g., Sekiya & Sudo, 2014; Sekiya et al., 2018). Moreover, CHASER is a forward model in the chemical reanalysis system (TCR) developed by Miyazaki et al. (2017, 2020). The model simulations are performed at a horizontal resolution of $2.8° \times 2.8°$ (T42). Although the model can run at

higher resolutions, T42 is the most commonly used framework for CHASER applications. Therefore, it is used for this study.

Hoque et al. (2022) validated CHASER-simulated $NO_2$ and HCHO against OMI and MAX-DOAS observations for 2017. CHASER showed good skills in reproducing the OMI- (spatial correlation, $r = 0.74$) and MAX-DOAS- (temporal correlation $R > 0.80$) observed HCHO abundances. The study found that biomass burning contributes ~50% to the HCHO levels observed at the site in Thailand. However, the limitations of the study are: (1) Simulated HCHO partial column and profile were evaluated against MAX-DOAS observation on a seasonal scale only, (2) Model sensitivity studies were site-specific, thus providing no global statistics on emission contribution, and (3) Satellite observations were used as supporting datasets; thus the model-satellite comparison has not been comprehensive. This study utilizes multi-satellite (TROPOMI and OMI) HCHO observations, different $NO_x$ emission inventories, aircraft measurements, and daily and diurnal MAX-DOAS data to provide robust and comprehensive statistics on the model HCHO simulations.

# 2 Model, observations, and methods

## 2.1 CHASER

CHASER 4.0 (ver. 4) is a global CTM that studies the atmospheric environment and radiative forcing. It is coupled online with the MIROC atmospheric general circulation model (AGCM) and the SPRINTAS aerosol transport model (Takemura et al., 2005, 2009). The latest version of CHASER (Ha et al., 2023; He et al., 2022) entails several updates, including the formation of aerosol species and related chemistry, radiation, and cloud processes.

Through 263 multi-phase (gaseous, aqueous, and heterogenous) chemical reactions, CHASER calculates the concentrations of 92 species considering the chemical cycle of $O_3$ – $NO_x$ (nitrogen oxides) – $HO_x$ (hydrogen oxides) -$CH_4$-CO along with oxidation of NMVOCs (Ha et al., 2023; He et al., 2022; Hoque et al., 2022; Miyazaki et al., 2017; Sekiya et al., 2023). The chemical mechanism is adopted mainly from the master chemical mechanism (MCM) (Jenkin et al., 2015). The stratospheric $O_3$ chemistry simulations are based on the Chapman mechanisms, the catalytic reaction of halogen oxides, and polar stratospheric clouds. The dry and wet depositions are calculated based on resistance-based parameterization (Wesley et al., 1984), cumulus convection, and large-scale condensation parameterization. Advective trace

transport is calculated using the piecewise parabolic method (Colella & Woodward, 1984) and flux-form semi-Lagrangian schemes. Tracer transport is simulated on a sub-grid scale in the framework of the prognostic Arakawa–Schubert cumulus convection scheme (Emori et al., 2001) and vertical diffusion scheme (Mellor & Yamada, 1974). The simulations were performed at a horizontal resolution of $2.8° \times 2.8°$, with 36 vertical layers from the surface to approx. 50 km altitude, with a 20 min time step. At every time step, meteorological fields obtained from the MIROC AGCM were nudged toward the 6-hourly NCEP FNL reanalysis data.

CHASER incorporates emissions from biomass burning, anthropogenic sources, lightning, and soil. Anthropogenic $NO_x$ emissions for 2018 are obtained from the HTAP_v3 inventory (Crippa et al., 2023). Other anthropogenic emissions are taken from the HTAPv2.2 for 2008 and the biomass burning emissions from MACC-GFAS (Inness et al., 2013). The monthly soil $NO_x$ emissions derived from Yienger and Levy (1995) are constant each year. Biogenic emissions of VOCs are obtained from a process-based biogeochemical model: the Vegetation Integrative Simulator for trace gases (VISIT) (Ito and Inatomi, 2012). VISIT is a part of the CHASER modeling framework and incorporates the biogenic flux estimate scheme of Guenther et al. (1997) (Ito et al., 2022). The global isoprene emissions in VISIT and CAMS global biogenic emission inventory (Sinderolova et al., 2022; based on MEGANv2.1) are 400 and 450 TgC/yr, respectively. Lightning $NO_x$ production estimates are based on the parameterization of Price and Rind (1992) and linked to the convection scheme of the AGCM. Global $NO_x$ emissions in CHASER are set to 43.80 TgN/yr considering industrial production (23.10 TgN /yr), biomass burning (9.65 TgN/yr), soil (5.50 TgN /yr), lightning (5 TgN/yr), and aircraft (0.55 TgN/yr) as significant emission sources. Annual monoterpene, acetone, and other non-methane volatile organic compound (ONMV) emissions are 102, 20, and 60 TgC/yr, respectively. Direct emissions of HCHO from anthropogenic sources and biomass burning are not considered in CHASER. However, secondary production of HCHO from VOCs ($C_2H_6$, $C_3H_8$, $C_2H_4$, $C_3H_6$, $CH_3COCH_3$, ONMV) emitted directly from anthropogenic and pyrogenic sources is considered.

Sekiya et al. (2018) comprehensively assessed CHASER simulated $NO_2$ abundances using OMI observations. CHASER well reproduced the ATom-observed OH spatiotemporal variation (Sekiya et al., 2018). The quality of $O_3$ simulations has been explained in the work of Sudo et al. (2014). Ha et al. (2023) and He et al. (2022) updated the heterogeneous chemistry and lightning $NO_x$ scheme, respectively. These updates have not been considered in the current study. The effect of these updates on the HCHO

simulations will be addressed in a separate study. Multiple simulations with varying emission inputs were
performed for the study. They are presented in Table 1.
**Table 1**. Combinations of emission inventories for different simulations used in this study

| Simulation name | NOx emissions | Biogenic emissions | Anthropogenic VOC emissions | Biomass burning |
|---|---|---|---|---|
| Standard | HTAP_v3 | ON | ON | ON |
| ANI[a] | HTAP_v3 | ON | Increased three-fold | ON |
| OLNE[b] | HTAP_v2.2 | ON | ON | ON |
| Biogenic_off | HTAP_v3 | OFF | ON | ON |
| Anthropogenic_off | HTAP_v3 | ON | OFF | ON |
| Biomass_off | HTAP_v3 | ON | ON | OFF |

[a] Anthropogenic VOC emission increased by three folds (ANI), [b]Simulation using old NOx emissions (OLNE)
To account for the altitude dependence of TROPOMI observations, averaging kernel (AK) information
obtained from the level (L2) files was applied to all simulations following the method of Sekiya et al.
(2018). First, the simulated HCHO profiles were sampled closest to the TROPOMI overpass of 13:30 LT
(Local Time). Secondly, AKs averaged on a 2.8° bin grid were applied to the sampled profiles. Then, the
total column was calculated. Thirdly, the AK-applied model columns on the available measurement days
were selected.
**2.2 TROPOMI**
The TROPOMI operational L2 offline (OFFL) HCHO vertical column density (VCD) (ver. 1.1.5.7) data
from 2019 to 2020 have been used for this study. The S5P TROPOMI HCHO L2 product user manual
(Veefkind et al., 2012) provides a detailed product description. The TROPOMI HCHO retrieval
algorithm is based on the DOAS technique, adapted directly from the OMI QA4ECV product retrieval
algorithm (De Smedt et al., 2017). The three-step retrieval algorithm was explained explicitly by De
Smedt et al. (2018). Slant columns were retrieved from the UV part of the spectra (Channel 3) in a 328.5–
358 nm fitting window. The HCHO cross-section data reported by Meller and Moortgart (2000) were
used to fit the spectra. All the cross-sections were convolved with the instrument slit function (adjusted
after the launch) for every row separately. Spectra averaged over the tropical Pacific region from the prior
day were used as reference spectra for the DOAS fit (De Smedt et al., 2021; Vigouroux et al., 2020). The

slant columns, therefore, exceed the average Pacific background HCHO levels because they were derived from the local and reference spectrum differences. The slant columns were converted to tropospheric columns (Nv) using a look-up table of vertically resolved air mass factors ($M$) at 340 nm calculated with the radiative transfer model VILDORT v2.6 (Spurr, 2008). The value of $M$ depends on the observation geometry, surface albedo, cloud properties, and a priori profiles of HCHO. The surface albedo at a spatial resolution of $1° \times 1°$ was extracted from the monthly OMI albedo climatology (Kleipool et al., 2008). Daily HCHO a  priori profiles were obtained from TM5-MP CTM at a similar spatial resolution. The independent pixel approximation (Boersma et al., 2004) approach was applied to pixels with cloud fractions greater than 0.1. Background correction was performed based on HCHO slant columns from the five prior days over the Pacific Ocean to account for any remaining global offsets and stripes (De Smedt et al., 2021). Background HCHO contribution from $CH_4$ oxidation in the reference region is calculated with TM5-MP. The resulting HCHO tropospheric column is calculated using equation (1):

$$N_v = \frac{N_s - N_{s,o}}{M} + \frac{M_o}{M} * N_{v,0}^{CTM} \tag{1}$$

where $M_o$ is the air mass factor of the reference sector. Following De Smedt et al. (2021), the following filters ensured the data quality: (1) cloud fraction less than 0.3, (2) quality assurance values greater than 0.5, (3) retrievals with solar zenith angle (SZA) less than 70°, (4) surface albedo less than 0.1, and (5) air mass factor greater than 0.1. The total uncertainty in the reprocessed TROPOMI HCHO columns was estimated as $>= 90\%$ for the fire-free region (Zhao et al., 2022, and references therein). The uncertainties in the air mass factors, slant column fitting, and background HCHO, respectively, account for 75, 25, and 40% of the total uncertainty. The estimated uncertainty in the retrievals in regions with strong fires is ~35%.  The filtering criteria of the TROPOMI datasets are as follows: quality assurance value (QA)>0.6, solar zenith angle <70º, cloud fraction < 0.3, AMF > 0.1, and surface reflectivity <0.2.

TROPOMI observations are averaged spatially and temporally to the CHASER grid (T42) daily, leading to horizontal representativeness errors. However, the random horizontal representativeness errors are in the order of 5-10%, which is lower than the individual retrieval error of the satellite observations (Boersma et al., 2015). If the model horizontal resolution is increased by 50% (i.e., simulated at a horizontal resolution of 1.4º × 1.4º), the change in HCHO abundances is less than 6% (Fig S1 and Table S1 in supplementary information). The vertical sensitivity of the satellite retrievals is the most relevant source of representativeness error (Boersma et al., 2015). The current study utilizes the  TROPOMI AK information to minimize the representativeness error. Therefore, the horizontal representative error will likely affect the results less than other error sources, such as uncertainties in satellite retrieval, emission inventories, and model chemical mechanisms.

**2.3 OMI**

 The comparison study used HCHO retrievals from OMI, a nadir-viewing spectrometer on board the Aura satellite, which measures backscattering solar radiation in the spectral range of 270–500 nm (Levelt et al., 2018). OMI crosses the equator at 13:40 LT (Zara et al., 2018) and provides daily global coverage of trace gases, including HCHO, at a spatial resolution of $13 \times 24$ km$^2$. For use in this study, HCHO columns

from 2019 to 2020, retrieved using the BIRA-IASBv14 (De Smedt et al., 2021), were obtained from the Aeronomie website (i.e., https://www.temis.nl/qa4ecv/hcho/hcho_omi.php, last accessed on 01/07/2023). The data-filtering criteria were cloud fraction < 0.3, SZA < 70°, quality flag =0, and cross-track quality flag = 0. Like TROPOMI, OMI data were also averaged spatially and temporally to the model grid(T42).

## 2.4 ATom-4 aircraft campaign

The NASA Atmospheric Tomography (ATom) mission used a DC-8 aircraft to study the remote atmosphere over the Pacific and Atlantic oceans from ~80° N to ~65° S (Wofsy et al., 2018). Repeated flights measured the vertical profiles from 0.15 to 12 km to provide information related to greenhouse gases, reactive and tracer species, and aerosol composition and size distribution (Kupc et al., 2018). Over two years and four phases, sampling was conducted in one of the four seasons in each stage (Zhao et al., 2022). Here, the 1-minute averaged measurements of HCHO and isoprene during the ATom-4 flight (Fig.S2) in 2018 are used for the model evaluation. The NASA In Situ Airborne Formaldehyde (ISAF) instrument (Cazorla et al., 2015) performed HCHO sampling based on the laser-induced fluorescence technique. Isoprene was measured using two instruments: (a) The University of Irvine Whole Air Sampler (WAS) and (b) the National Center for Atmospheric Research (NCAR) Trace Organic Gas Analyzer (TOGA). WAS sampled the air every 3–5 min, with subsequent analyses in the laboratory using gas chromatography (Simpson et al., 2020). TOGO sampling was conducted every 2 min with a 35 s integrated sampling time (Apel et al., 2021). The uncertainty in the WAS and TOGA isoprene observations are, respectively, ±10 and 15%. Measurement uncertainties in HCHO were reported as 10%. The simulations have been interpolated to the observed spatial and temporal resolution following the method of He et al. (2022). The observed and interpolated HCHO and isoprene vertical profiles were averaged over a 300-meter bin. The Atom campaign took place between 2016 and 2018.

## 2.5 MAX-DOAS observations

HCHO columns and the volume mixing ratio (vmr) were retrieved from two-year (2019–2020) MAX-DOAS observations at Phimai (15.18°N, 102.46°E, 212 m a.s.l.), Chiba (35.62°N, 140.10°E, 21 m a.s.l.), and Kasuga (33.52°N, 130.47°E, 28 m a.s.l.). The MAX-DOAS observations were conducted under the framework of the international air quality and sky research remote sensing (A-SKY) network (Irie, 2021). The sites were selected because continuous measurements from 2019 to 2020 were available for these sites. Phimai is a rural site in Thailand and experiences biomass burning influence from January to April. The climate is divided into two seasons- (1) dry season (January to May) and (2) wet season (June to December). Chiba and Kasuga are urban sites in central and southern Japan, respectively. The seasonal classification at these sites is – Spring (March to May), Summer (June to August), Autumn (September to November), and winter (December to February). The observations at these sites are described elsewhere (i.e., Hoque et al., 2018a; Irie et al., 2011,2015).

The A-SKY MAX-DOAS system, including the instrument and algorithm, participated in the Cabauw Intercomparison campaign for Nitrogen Dioxide measuring Instruments (CINDI) and CINDI-2 (Kreher et al., 2020; Roscoe et al., 2010) campaigns. The instrumentation has been described explicitly by Irie et al. (2008, 2011, 2015). A UV spectrometer (Maya2000Pro; Ocean Insight, Inc.) recorded high-resolution

spectra from 310–515 nm at six elevation angles (ELs) of 2°, 3°, 4°, 6°, 8° and 70°, which were repeated every 15 min. The reference spectra were recorded at EL of 70° instead of 90° to avoid saturation intensity. Spectra measured at all ELs were considered in the retrieved vertical profile and total columns. Consequently, the choice of reference ELs has no appreciable effect on the retrieval. The systematic error in the oxygen collision complex ($O_4$) was reduced by limiting the off-axis ELs to less than 10° (Irie et al., 2015). However, this limitation reduces sensitivity above the planetary boundary layer (PBL), maintaining high sensitivity in the lower layers of the retrieved profiles. The high-resolution solar spectrum measured by Kurucz et al. (1984) was used for daily wavelength calibration. The spectral resolution is approximately 0.4 nm at 357 and 476 nm (Hoque et al., 2022). Aerosol and trace gas columns and profiles were retrieved using the Japanese vertical profile retrieval algorithm JM2 (ver. 2) (Irie et al., 2011, 2015). Three-step profile and column retrievals by JM2 are explained explicitly in earlier reports (e.g., Hoque et al., 2018; Irie et al., 2011, 2015). The partial VCD values are converted to the volume mixing ratio (vmr) by scaling the U.S. standard atmosphere temperature and pressure data to the respective site surface measurements. The estimated total error (random and systematic) in the HCHO product is 30% (Hoque et al., 2022). Following Irie et al. (2011) and Hoque et al. (2018a, 2022), cloud screening was performed to ensure data quality.

# 3 Results and discussion

## 3.1 Comparison of CHASER HCHO with TROPOMI observations

Figure 1 presents a comparison of global distributions of annual mean HCHO columns obtained from TROPOMI retrievals and standard CHASER simulations at the TROPOMI overpass time (13:30). Differences between the observations and model simulations in the respective years are also depicted. The statistics related to the comparison are presented in Table 2. The simulation results agree well with the TROPOMI observations, with a global spatial correlation ($r$) of 0.93, mean bias error (MBE) (CHASER–TROPOMI) of -0.20 × $10^{15}$ molecules cm$^{-2}$, and root-mean-square error (RMSE) of 0.75 × $10^{15}$ molecules cm$^{-2}$. The $r$, MBE, and RMSE values between 60° S and 60° N were, respectively, 0.92, 0.13 × $10^{15}$ molecules cm$^{-2}$, and 0.82 × $10^{15}$ molecules cm$^{-2}$. CHASER HCHO columns are negatively biased relative to the TROPOMI retrievals. Table S2 shows the MBE and RMSE values obtained for the individual months. No seasonal variation in the systematic differences was observed between CHASER and TROPOMI. Biases can originate from uncertainties in the retrieval and model assumptions. TROPOMI HCHO retrievals greater than 8 × $10^{15}$ molecules cm$^{-2}$ were negatively biased by 25% relative to the ground-based MAX-DOAS observations (De Smedt et al., 2021), whereas direct emissions of HCHO were not considered in CHASER.

TROPOMI and CHASER show high HCHO concentrations over South America, central Africa, India,
eastern China, and Southeast Asia. Simulated HCHO magnitudes in the hotspot regions were 0.8–1.4 ×
$10^{16}$ molecules cm$^{-2}$, slightly higher than the observed range of 0.8–1 × $10^{16}$ molecules cm$^{-2}$. The dataset's
greatest differences (~4 × $10^{15}$ molecules cm$^{-2}$) were observed over Brazil and Southeast Asia. The
datasets show strong congruence in the high-latitude regions. The simulated and observed HCHO
columns over Europe, the Middle East, Japan, and Russia were 0.3–0.6 × $10^{16}$ molecules cm$^{-2}$. Simulated
HCHO columns (~3 × $10^{15}$ molecules cm$^{-2}$) over the remote Pacific region were consistent with the
observations, too. The remote Pacific regions represent background conditions strongly linked to $CH_4$
oxidation. Congruence with observations in this region suggests that the simulated $CH_4$ estimates in the
remote areas are reasonable.



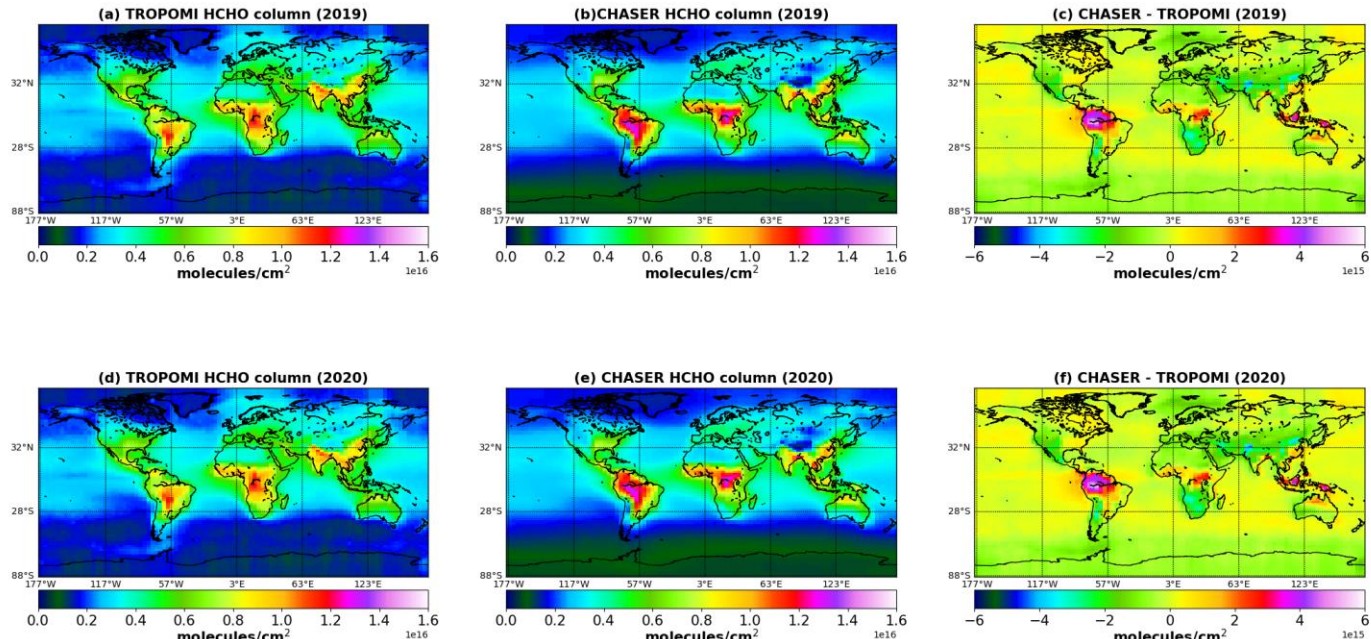

**Figure 1.** Annual mean HCHO columns ($\times 10^{16}$ molecules cm$^{-2}$) in 2019 and 2020 were obtained from TROPOMI retrievals (first column) and standard CHASER simulation (second column). The differences between the model and observations in the respective years are shown in the third column. The unit of difference is $\times 10^{15}$ molecules cm$^{-2}$.

**Table 2.** Comparison of annual mean HCHO ($\times 10^{16}$ molecules cm$^{-2}$) column between TROPOMI retrievals and CHASER simulations in 2019 and 2020. MBE and RMSE are the abbreviated forms of mean bias error and root mean square error, respectively. Units of MBE and RMSE are $\times 10^{15}$ molecules cm$^{-2}$. Correlation signifies the spatial correlation between the datasets.

| Year | Correlation | MBE | RMSE |
|------|-------------|-------|------|
| **2019** | 0.93 | -0.20 | 0.75 |
| **2020** | 0.93 | -0.19 | 0.75 |

304

305

306

Figure 2 compares the observed and simulated seasonality in HCHO columns ($\times$ 10$^{16}$ molecules cm$^{-2)}$) in different regions. Datasets for 2019 and 2020 were used to calculate the observed and simulated monthly mean values. The MBE ($\times$ 10$^{15}$ molecules cm$^{-2}$) between TROPOMI and CHASER HCHO columns in each region is shown in blue. The comparison statistics are given in Table 3. The regional boundaries are shown on the global distribution map in Fig. S3. Temporal correlations derived from daily values over two years are provided in Table S2.

313

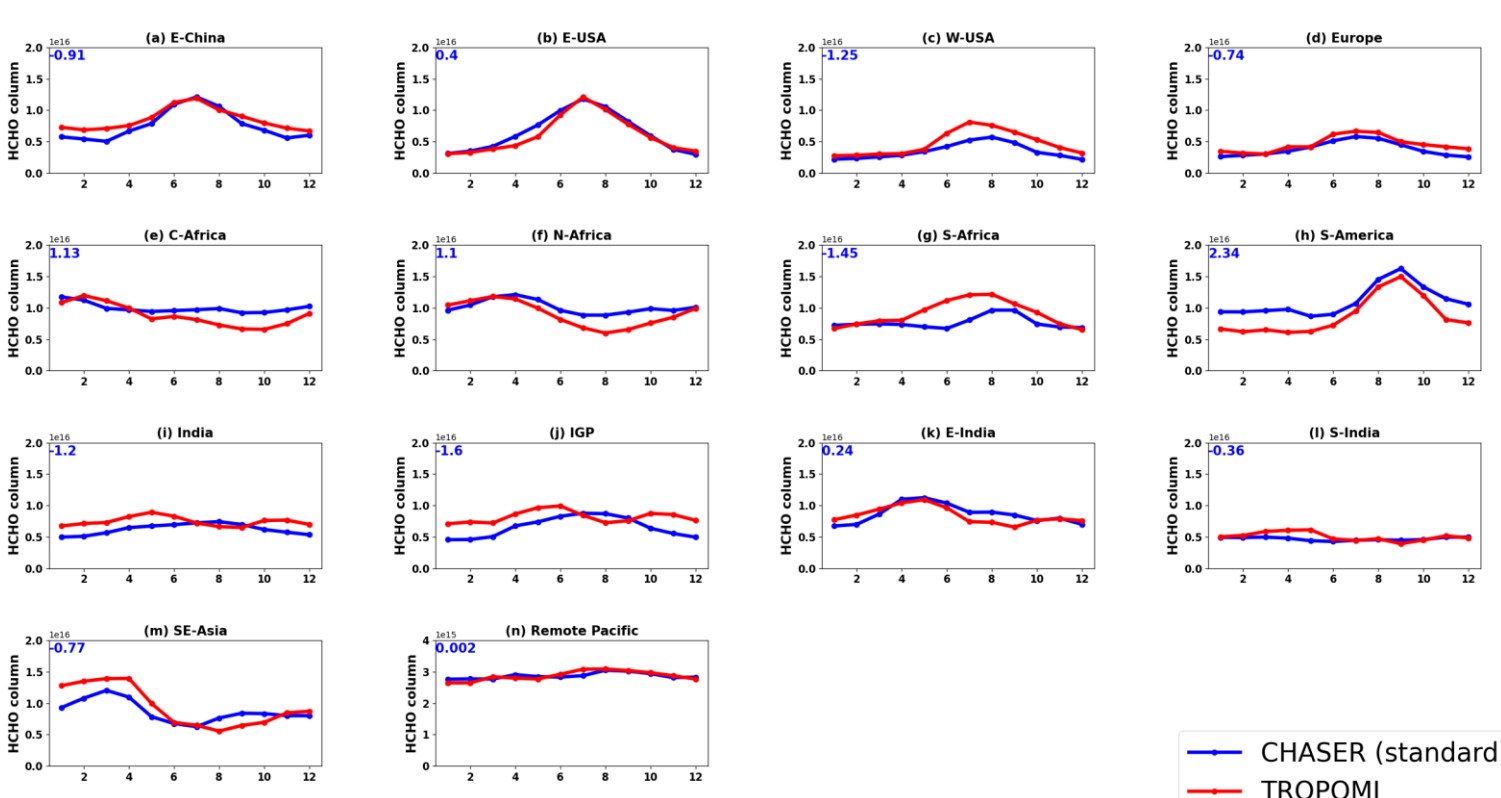

314

**Figure 2.** Seasonal variation in HCHO columns ($\times$ 10$^{16}$ molecules cm$^{-2}$) in eastern (a) China (E-China; 30–40°N, 110–123°E), (b) eastern United States (E-USA; 32–43°N,95–71°W), (c) western United States

(W-USA; 32–43°N, 125–100°W), (d) Europe (35–60°N, -10°W–30°E), (e) central Africa (C-Africa; 4°S-5°N, 10° – 40°E), (f) northern Africa (N-Africa; 5–15°N, 10°W–30°E), (g) southern Africa (S-Africa; 5–15°S, 10–30°E), (h) South America (S-America; 20°S – 0°N, 50–70° W), (i) India (7.5–35°N, 68–89°E), (j) the Indo Gangetic Plain (IGP; 21–33°N, 72–89°E), (k) east India (E-India; 15–25°N, 80–90°E)), (l) south India (S-India; 0–15°N, 63–80°E), (m) Southeast Asia (SE-Asia, 10–20°N, 96–105°E), and (n) the remote Pacific region (28°S – 32°N, 117°–177°W) as inferred from CHASER simulations (blue) and TROPOMI observations (red). Blue numbers denote MBE between the TROPOMI and CHASER HCHO columns. The observed and simulated mean values represent the average of 2019 and 2020.

### (a) E-China

Over E-China (Fig.2(a)), the datasets are moderately correlated spatially ($r$ =0.44), with MBE and RMSE values of -0.9 and $1.62 \times 10^{15}$ molecules cm$^{-2}$, respectively. The simulated seasonality correlates strongly with the observations ($R$= 0.97), with a consistent peak ($1 \times 10^{16}$ molecules cm$^{-2}$) in the HCHO variability in July. The HCHO columns' peaks are compatible with the peak in isoprene concentrations (Fig. S4), manifesting a strong biogenic contribution during summer. CHASER mostly underestimated the winter-time HCHO columns in this region. Liu et al. (2021) reported vehicular exhaust, solvent usage, and combustion-related regional transport as the primary VOC emission sources during winter in Shanghai, a megacity in eastern China. NMVOC emissions from these sources (i.e., vehicular exhaust, solvent usage, and transport) are considered in the HTAPv2.2 inventory (Crippa et al., 2023). Although CHASER considered HCHO production from the degradation of anthropogenic VOCs, it is likely underestimated, resulting in a lower simulated winter-time HCHO column in this region.

**Table 3:** Comparison of monthly mean tropospheric HCHO ($\times 10^{16}$ molecules cm$^{-2}$) columns obtained from TROPOMI retrievals and standard CHASER simulations. Coincident dates in 2019 and 2020 are used to calculate the statistics. Units of MBE and RMSE are $\times 10^{15}$ molecules cm$^{-2}$. The temporal correlations are derived from the seasonal means.

| Region | MBE (model – TROPOMI) | RMSE (model – TROPOMI) | Spatial Correlation (r-value) | Temporal Correlation (R-value) |
| --- | --- | --- | --- | --- |
| E-China | -0.91 | 1.62 | 0.44 | 0.97 |
| E – USA | 0.40 | 0.43 | 0.97 | 0.97 |
| W-USA | -1.25 | 1.29 | 0.85 | 0.95 |
| Europe | -0.74 | 0.92 | 0.73 | 0.93 |
| C-Africa | 1.13 | 1.52 | 0.93 | 0.74 |
| N-Africa | 1.10 | 1.26 | 0.87 | 0.83 |
| S-Africa | -1.45 | 1.64 | 0.89 | 0.59 |
| S-America | 2.34 | 2.85 | 0.56 | 0.97 |
| India | -1.20 | 1.77 | 0.84 | 0.18 |
| IGP | -1.60 | 1.99 | 0.91 | 0.44 |
| E-India | 0.24 | 1.08 | 0.86 | 0.72 |
| S-India | -0.36 | 0.52 | 0.96 | 0.34 |
| SE-Asia | -0.77 | 1.22 | 0.71 | 0.87 |
| Remote Pacific | 0.002 | 0.13 | 0.86 | 0.76 |

**(b) Eastern USA, western USA, and Europe**

CHASER has well-reproduced the HCHO spatial variability in the eastern USA (E-USA; Fig.2(b); $r=0.97$) and western USA (W-USA; Fig.2(c); $r=0.85$). The peaks in the HCHO variability coincide with the isoprene peak in these regions (Fig. S4). The simulated amplitude of the HCHO seasonal modulation in E-USA and W-USA are 74 and 62%, whereas the observed seasonal amplitudes are 74 and 65%, respectively. The peak in the HCHO seasonality in E-USA is similar in both datasets ($\sim1.2 \times 10^{16}$ molecules $cm^{-2}$). The RMSE value in the W-USA region is 15% higher than in E-USA. Although the spatial correlation in Europe (Fig.2(d)) is moderate ($r=0.73$), the temporal correlation is strong ($R=0.95$). The simulated and observed HCHO seasonal modulations in Europe are 60% and 62%, respectively. The model–satellite discrepancies are prominent in Europe and W-USA during summer and autumn. In both regions (i.e., Europe and W-USA), the biogenic and anthropogenic contribution to the total HCHO level is equivalent during summer. In autumn, the anthropogenic emission contributions are higher. (Section 3.8). This manifests a potential model underestimation of biogenic HCHO levels in these regions, linked to the uncertainties in the biogenic emission inventory and isoprene mechanism. However, the model–satellite agreement is strong during the winter in these regions. During winter, anthropogenic VOC emissions drive the HCHO variability in these regions (Luecken et al., 2018; Pozzani et al., 2002). Therefore, the simulated contribution of anthropogenic sources to the HCHO abundances during winter in these regions is reasonable.

**(c) Central, Northern, and Southern Africa**

Over the African regions (Fig.2 (e-g), the spatial correlation is higher than 0.80. The African continent is the single largest biomass-burning emission source (Roberts et al., 2009). The observed and simulated amplitude of the HCHO seasonality in central Africa (C-Africa; Fig.2(e)) are, respectively, 45 and 21%. The mean simulated and observed HCHO abundances in North Africa (N-Africa; Fig.2(f))) during the biomass burning season is $\sim1.06 \times 10^{16}$ molecules $cm^{-2}$, consistent with the GOME-2 and SCIAMACHY observations (De Smedt et al., 2008). Figure S5 (Supplementary Information) shows the seasonal fire radiative power (FRP) cycle over the three African regions. FRP, a measure of outgoing radiant heat from fires, is a tracer of changes in atmospheric trace constituents related to pyrogenic emissions (Hoque et al.,

2018a). The observed and simulated enhanced HCHO columns in N-Africa are congruent with the high FRP values, manifesting the contribution of biomass burning to the HCHO abundances. CHASER could not replicate the observed HCHO seasonality over C-Africa. However, simulations show a decrease in the HCHO abundances in C-Africa from January to March, consistent with the changes in the coincident FRP values.

Over South Africa (S-Africa; Fig.2(g)), elevated TROPOMI HCHO columns are consistent with GOME-2 and SCIAMACHY observations (De Smedt et al., 2008). The observed peaks in HCHO columns and FRP values (Fig.S5) are consistent and thus can be attributed to biomass burning. Pyrogenic emissions contribute ~36% to the high HCHO columns in this region (section 3.8). TROPOMI and CHASER have captured the shift in biomass-burning seasons from northern to southern Africa, which agrees well with earlier observations (i.e., GOME-2, SCIAMACHY). The observed amplitude of the HCHO seasonal cycle in South and North Africa is 46%, signifying an almost two-fold increase in HCHO abundances during the biomass-burning season. Earlier studies (e.g., De Smedt et al., 2008; Muller et al., 2008) found that such a feature (increment by a factor of 2) exists only in the Southern African region. This likely indicates an increase in fire intensity in Northern Africa.

**(d) South America**

CHASER showed moderate skill in reproducing the observed HCHO spatial distribution in South America (S-America; Fig 2(h); $r = 0.56$). However, the seasonal variation in the HCHO columns is strongly correlated ($R = 0.97$). The MBE and RMSE in the South American continent are $2.34 \times 10^{15}$ and $2.385 \times 10^{15}$ molecules cm$^{-2}$, respectively. The enhanced HCHO columns during the South American biomass burning season are well reflected in the datasets. They show a distinctive seasonal cycle. The observed and simulated mean HCHO columns from August through October are ~$1.5 \times 10^{16}$ molecules cm$^{-2}$. CHASER estimated 46% seasonal modulation in the HCHO abundances, whereas the observed modulation is 59%. The model overestimates the HCHO columns in S-America, similarly to C-Africa and N-Africa, probably because of the uncertainties in biogenic emission inventories and the isoprene oxidation scheme.

**(e) India**

CHASER well reproduced the observed HCHO spatial distribution in India ( Fig.2 (i); $r$ =0.84), with MBE and RMSE of $-1.20 \times 10^{15}$ and $1.775 \times 10^{15}$ molecules cm$^{-2}$. However, the temporal correlation ($R$=0.18) between the datasets is low. The observed seasonal modulation of ~30% manifests a less-prominent seasonality in HCHO abundances in India. The correlation between temperature variations and isoprene emissions in India is inhomogeneous (Starvakou et al., 2014). India has a diverse landscape, including major forests over the east, northeast, and southwest regions and deserts in northwestern India (Surl et al., 2018). The Indo-Gangetic Plain (IGP) stretches from Eastern Pakistan to Bangladesh and is a major agricultural region in India (Kuttippurath et al., 2022). Thus, averaging the HCHO columns over a diverse landscape can lead to a less prominent seasonality. Moreover, biomass burning compromises 23% of India's total NMVOC (13 Tg/yr)  emissions (Stewart et al., 2021). Sensitivity analysis (section 3.8) estimates show biomass burning contribution to the HCHO levels in India is ~2%., manifesting that the modeled biomass burning emissions for India are underestimated. Considering the diverse Indian landscape, the model satellite comparison over three regions in India (IGP, east India, and South India) is shown in Fig.2 (j-l).

The model has shown good skill in reproducing the observed HCHO spatial variation in the IGP (Indo-Gangetic Plain; Fig.2(j)) region ($r$ = 0.91). However, the temporal correlation is moderate ($R$=0.44). Several field studies (e.g., Hoque et al., 2018b) have reported biomass-burning influences during spring and autumn in IGP, explaining the elevated observed HCHO columns. HCHO seasonal variation during January–June is consistent in both datasets, with an $R$-value of 0.78. The mean observed and modeled HCHO abundances during spring in IGP are, respectively, $1.19 \times 10^{16}$ and $8.72 \times 10^{15}$ molecules cm$^{-2}$. However, the model could not reproduce the autumn-time biomass-burning events, reducing the overall $R$-value in the IGP region. CHASER underestimates winter HCHO columns in the IGP region. Liquid petroleum gas (LPG) usage, evaporative fuels, and garbage burning contribute significantly to winter NMVOC levels in Delhi and Mohali (Kumar et al., 2021). Although NMVOC emissions from these sources are considered in the simulations, they are likely underestimated in the IGP region.

Over East India (Fig.2(k)), both the spatial ($r = 0.86$) and temporal ($R = 0.72$) agreement between
TROPOMI and CHASER HCHO are strong. The observed and modeled amplitudes of the HCHO
seasonal cycle are 40%. Both datasets show enhanced HCHO levels during spring., consistent with high
isoprene concentrations (Fig.) Biogenic emissions are the main driver of the HCHO levels in East India;
however, emissions from mines are also potential sources of $NO_x$ and VOCs (Kuttippurath et al., 2022).

Similarly, CHASER has shown a strong capability for reproducing the HCHO spatial distribution
($r=0.96$) in south India (S-India; Fig..2(l)). However, the temporal correlation is low. The mean observed
and simulated HCHO abundances are, respectively, $4.68 \times 10^{15}$ and $5.03 \times 10^{15}$ molecules cm$^{-2}$. The
HCHO seasonality in S-India is less prominent than in the other two regions. The coordinates bounds
defined for S-India in this study compromise a large portion of the southern coastal region, which
experiences a tropical maritime climate with limited seasonal variations in temperature (Surl et al., 2018).
Such a feature can potentially lead to a less prominent HCHO seasonality in S-India.


**(f) Southeast Asia**
In Southeast Asia (SE-Asia; Fig.2(m)), the $r$-value is 0.71. The MBE and RMSE are respectively $-0.77 \times$
$10^{15}$ and $1.2 \times 10^{15}$ molecules cm$^{-2}$. During the dry season (January–April), prominent biomass burning
occurs in this region in many countries (e.g., Thailand, Malaysia, Indonesia, Cambodia). Such fire events
degrade local air quality and cause transboundary pollution (Hoque et al., 2018; Kahn et al., 2016).
TROPOMI and CHASER have well-captured the pyrogenic emissions-led enhanced HCHO levels. The
simulated and observed mean dry season HCHO columns are, respectively, $1.07 \times 10^{16}$ and $1.35 \times 10^{16}$
molecules cm$^{-2}$. The observed and simulated amplitude of the seasonal cycle are, respectively, 48 and
60%. CHASER-reproduced columns during the dry season are underestimated. Potential reasons for such
discrepancies are discussed in section 3.3.

**(g) Remote Pacific region**

The datasets correlate strongly over the remote Pacific region (Fig.2 (n)), representing the background condition. No prominent seasonal variation is observed in this region, which CHASER has well simulated. The simulated and observed background HCHO column is $2.86 \times 10^{15}$ molecules cm$^{-2}$.


## 3.2 Comparison over countries with large forested areas

Figure 3 shows the observed and simulated HCHO columns over countries where large forested areas are located. The definition of the countries is adopted from the work of Opacka et al. (2021). The statistics presented in Table 4 include regions with high and low biogenic activities. This section compares the overall biogenic emissions in the defined regions with literature values and assesses their impact on model performance.

Over China, CHASER correlates strongly with TROPOMI ($r = 0.92$), with MBE of $-3 \times 10^{15}$ molecules cm$^{-2}$. The lowest differences between the datasets are observed primarily in the southeastern and western parts of China. Shanghai, Nanjing, and Guangzhou megacities are located in southeastern China. Consequently, CHASER has demonstrated good skills in the areas encompassed by multiple megacities. The annual isoprene emission for China in CHASER is 34 TgC/yr, higher than that of Opacka et al. (2021) (9.5–23 TgC/yr).

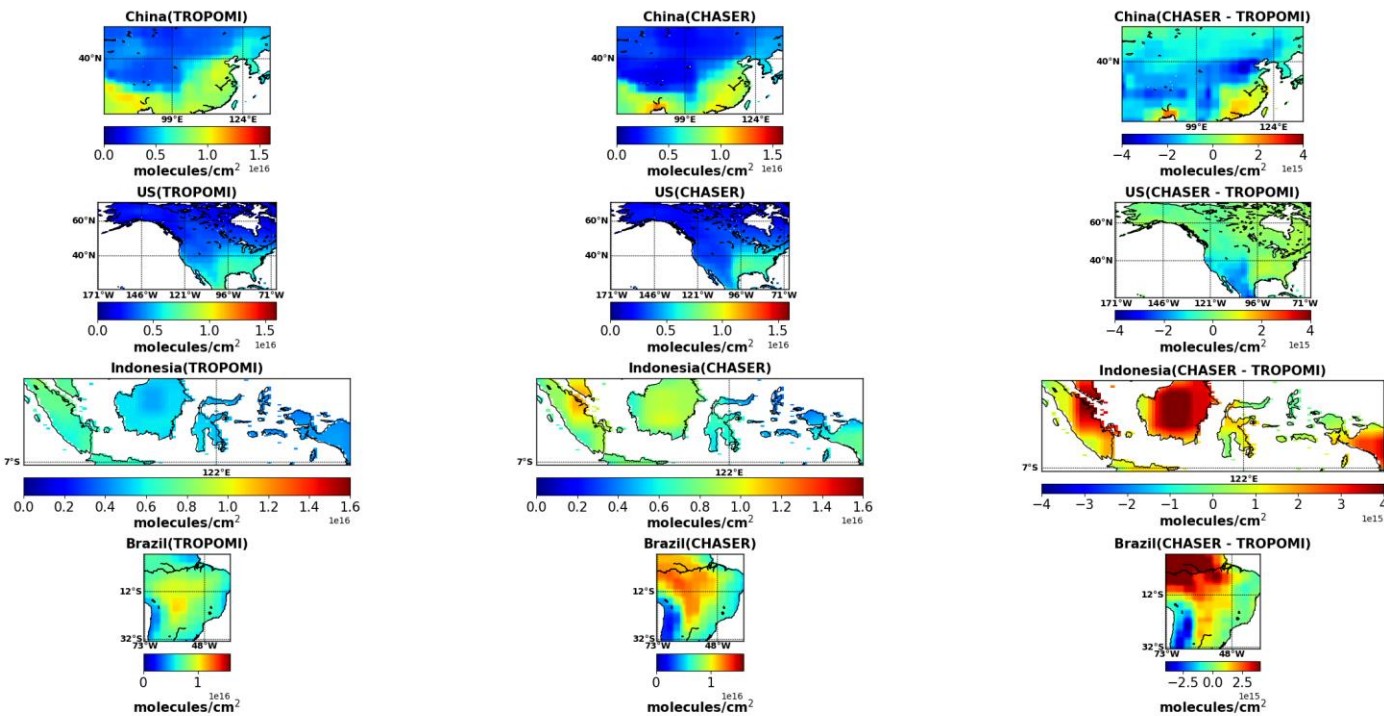

**Figure 3:** Two-year (2019 and 2020) mean CHASER (first column) and TROPOMI (second column) HCHO columns ($\times 10^{16}$ molecules cm$^{-2}$ cm$^{-2}$) in China (18.19–53.45°N, 73.67–135.02°E), United States (18.91–45°N, 66–171°W), Indonesia (10°S–6°N, 95–142°E), and Brazil (33°S – 5.24°N, 34–73°W). The differences between the datasets are presented in the third column. Only the coincident dates among the datasets are used to calculate the annual mean data.

**Table 4**: Comparison of two-year mean HCHO ($\times 10^{15}$ molecules cm$^{-2}$) column between TROPOMI and CHASER over countries with large forested areas. The coordinate bounds of the regions are adapted from

Opacka et al. (2020). Correlation signifies the spatial agreement between CHASER and TROPOMI,
calculated from the annual mean data. The unit of MBE is $\times 10^{15}$ molecules cm$^{-2}$

| Region | Spatial correlation (model vs. TROPOMI) | MBE (model–TROPOMI) |
|---|---|---|
| China | 0.92 | -0.84 |
| US | 0.93 | -0.05 |
| Indonesia | 0.81 | 1.05 |
| Brazil | 0.84 | 1.06 |



CHASER has shown excellent skill in reproducing TROPOMI observations over the US. Along with high
*r*-values, the simulated magnitude of the HCHO columns is consistent with observations throughout the
whole region. Consequently, the bias between the datasets for the US is 2%. In CHASER, annual isoprene
emissions in the US and the southeastern US are 22 and 7.8 TgC/yr, respectively. Such values are within
the ranges reported by Stavrakou et al. (2015) and Opacka et al. (2021).

The MBE between TROPOMI and CHASER in Indonesia is $1.05 \times 10^{15}$ molecules cm$^{-2}$. The *r*-value is
0.81. Indonesia's annual mean TROPOMI and CHASER HCHO abundance is $5.06 \times 10^{15}$ and $6.15 \times 10^{15}$
molecules cm$^{-2}$. The most significant differences between the datasets ($4 \times 10^{15}$ molecules cm$^{-2}$) are
observed for Sumatra, Borneo, and Sulawesi islands. Annual isoprene emissions in Indonesia used in the
CHASER simulations are 42 TgC/yr. Indonesian isoprene emissions vary between 25.5 to 32 TgC/yr
depending on the land-use change (Opacka et al., 2021). Top-down estimates based on OMI and GOME-
2 observations are ~11 TgC/yr (Stavrakou et al., 2015). However, the 11 TgC/yr emissions are half of the
top-down estimates based on SCIAMACHY observations. Consequently, isoprene emissions in Indonesia
remain largely uncertain. However, CHASER estimates with the VISIT emissions are higher than those
reported in the literature, likely leading to the model overestimation in Indonesia.

CHASER overestimates the HCHO columns over the Amazonia, mostly in northern Brazil. Fig.S6 shows
the observed and simulated seasonal HCHO variation over Brazil. Although the model reproduced the
temporal variability well, the magnitude was overestimated. This indicates that emission uncertainties are
more prominent than uncertainties related to the chemical mechanism for this region. In CHASER, annual
isoprene emissions over Amazonia are 67 Tg/yr, consistent with the OMI-based top-down estimates of
70 Tg/yr, estimated using apriori emissions from MEGAN (Stavrakou et al., 2015). However,
deforestation affects the VOC emissions in the Amazon (Yáñez-Serrano et al., 2020). Massive
deforestation in the Amazon occurred between 1985 and 2020, changing 11% of the Amazonian biome
(Cabarello et al., 2022). Depending on the land use and land cover change(LULCC), isoprene emissions
in Brazil can vary between 79. And 106.5 Tg/yr (Opacka et al., 2021). Moreover, although biogenic VOC
modeling in the Amazon has improved, VOC dynamics in the changing Amazonian biome are poorly
understood (Salzar et al., 2018; Taylor et al., 2018). Therefore, updated biogenic VOC and LULCC
inventories can potentially improve the model performance in Brazil.
In addition, CHASER isoprene emission estimates for Europe and Russia are, respectively, 17 and 15
TgC/yr, which are comparable to values reported in the literature (e.g., Guenther et al., 2006; Sinderolova
et al., 2022).
The discussion is based on isoprene emissions because isoprene is the dominant biogenic VOC (BVOC).
Although not included in the current discussion, the chemical yield of HCHO from the oxidation of other
BVOCs might also be a source of model uncertainty.


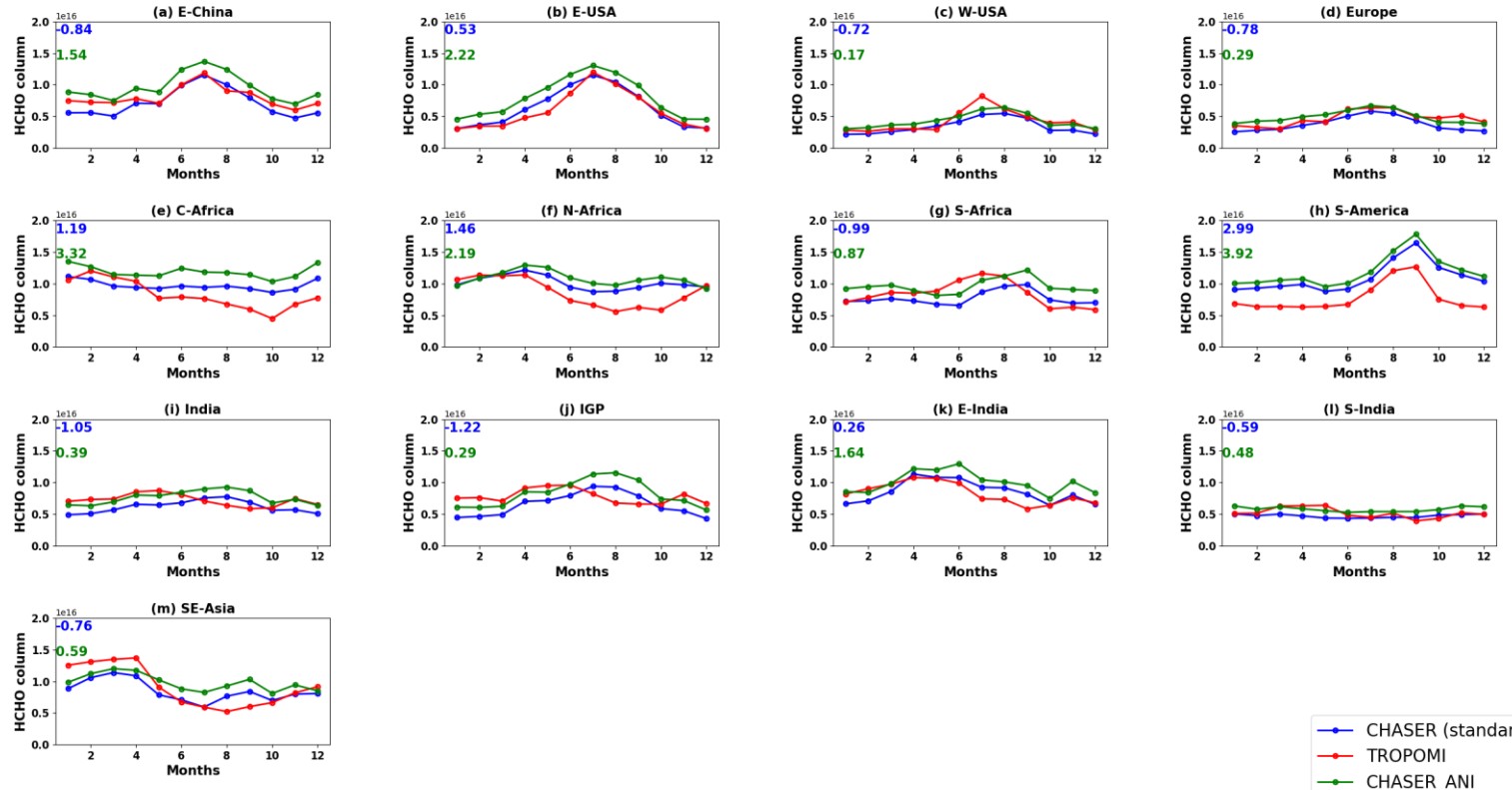

**Figure 4:** Seasonal variation of HCHO ($\times 10^{16}$ molecules cm$^{-2}$) in the selected regions, as inferred from standard simulations (blue), TROPOMI observations (red), and ANI estimate (green). Anthropogenic VOC emissions are increased threefold in the ANI simulations. The blue numbers denote MBE between the TROPOMI and CHASER HCHO columns. The MBE between the ANI and TROPOMI columns is shown in green. The coordinate bounds of the regions are similar to those in Fig. 2. Simulations and observations in 2019 were used to calculate the monthly mean values.

## 3.3 Uncertainties related to anthropogenic VOC emissions

Uncertainties in anthropogenic VOC emissions can also be crucially important. Sensitivity simulations are performed by perturbing the anthropogenic VOC emissions. Perturbation effects are relevant when the anthropogenic VOC emissions are increased by threefold or more. We select the lowest perturbed simulation (i.e., threefold increase; hereafter ANI). A better agreement between ANI and TROPOMI HCHO columns is attributed to underestimated anthropogenic VOC emissions in the standard simulation.

Figure 4 compares the TROPOMI HCHO columns and ANI simulations in 2019. Standard simulation
estimates for 2019 are also shown. The comparison statistics are provided in Table 5.
Over E-China (Fig.4(a)) and India (Fig.4(i)), ANI shows better agreement with TROPOMI than the
standard simulation during winter. In India and China, the contribution of anthropogenic emissions to the
NMVOC levels is more significant during the winter (Kumar et al., 2021; Liu et al., 2021). Thus, the ANI
simulations improve the contribution of the winter-time anthropogenic VOCs in these regions. The ANI
MBE and RMSE values over E-China are higher than the standard simulation. This indicates the
anthropogenic VOC estimates in E-China during the other seasons are reasonable. In contrast, the ANI
simulations reduce the MBE values in India, manifesting a higher underestimation of anthropogenic VOC
emissions in this region than in E-China.
Similar to E-China, the ANI MBE and RMSE values are higher in C-Africa, N-Africa, S-Africa, South
America, and E-USA. Over Europe (Fig.4(d)) and W-USA(Fig.4(c)), ANI RMSE values are lower than
the standard simulation. The ANI simulations replicated the observed HCHO column magnitude in both
regions from October to December, resulting in lower RMSE values.
ANI estimates during the dry season in SE Asia (Fig.4(m)) are similar to the standard simulation values,
indicating a small effect of anthropogenic emission uncertainties. The dry season columns are
overestimated when the anthropogenic VOC emissions are increased fivefold (Fig. S7). Space-based
observations have provided substantial evidence of increasing anthropogenic VOC emissions in Asian
cities (Bauwens et al., 2022). Therefore, the anthropogenic VOC emission inventory should be updated
to reduce the discrepancy between CHASER and TROPOMI over SE-Asia.

**Table 5:** Comparison among regional mean tropospheric HCHO ($\times 10^{16}$ molecules cm$^{-2}$) columns
inferred from TROPOMI observations, standard simulation, and ANI estimates. Units of MBE1, MBE2,
RMSE1, and RMSE 2 are $\times 10^{15}$ molecules cm$^{-2}$. The simulations and observations for 2019 were used
to calculate the statistics.


| Region | MBE1 (Standard–TROPOMI) | MBE2 (ANI–TROPOMI) | RMSE1 (Standard–TROPOMI) | RMSE2(ANI–TROPOMI) |
|---|---|---|---|---|
| E-China | -0.84 | 1.54 | 1.40 | 1.74 |
| E-USA | 0.53 | 2.22 | 0.58 | 2.25 |
| W-USA | -0.72 | 0.17 | 0.80 | 0.43 |
| Europe | -0.78 | 0.29 | 0.92 | 0.67 |
| C-Africa | 1.19 | 3.32 | 1.57 | 3.60 |
| N-Africa | 1.46 | 2.19 | 1.61 | 2.30 |
| S-Africa | -0.99 | 0.87 | 1.32 | 1.39 |
| S-America | 2.99 | 3.92 | 3.41 | 4.28 |
| India | -1.05 | 0.39 | 1.57 | 1.50 |
| IGP | -1.22 | 0.29 | 1.69 | 2.02 |
| E-India | 0.26 | 1.64 | 1.22 | 2.11 |
| S-India | -0.59 | 0.48 | 0.69 | 0.58 |
| SE-Asia | -0.76 | 0.59 | 1.16 | 0.78 |

## 3.4 Impacts of NO$_x$ emissions uncertainties on HCHO simulations

Uncertainties in the NO$_x$ emissions can affect the HCHO abundances through the NO$_x$-HO$_x$-VOC cycle. Such effects are assessed by comparing simulations with different NO$_x$ inventories with the TROPOMI observations. The CHASER standard, OLNE, and TROPOMI HCHO columns are depicted in Fig. 5. The HTAP_v3 NO$_x$ emission inventory is replaced with the HTAP_v2.2 inventory in the OLNE simulations without altering the remaining emission inventories. The differences between the two NO$_x$ inventories are – (1) HTAP-v3 inventory considers the changes in NO$_x$ emissions from 2000 to 2018, whereas the temporal coverage of HTAP_v2.2 is 2008 – 2010, and (2) Emissions in HTAP-v3 have a higher sectoral disaggregation (Crippa et al., 2023). The comparison-related statistics are given in Table S3. NO$_x$ emissions from both inventories are shown in Fig. S8

On a global scale, HCHO column estimates are mostly unaffected by the changes in the NO$_x$ emission inventories, manifested by the MBE values (Table 6). However, RMSE is 8% lower in the case of standard simulation. OLNE estimates in the higher latitude (>=50ºN) are 5% lower than the standard simulations. Such differences do not affect the model–satellite agreement in these regions.

The standard HCHO columns in India, China, and Southeast Asia are approximately 10–20% lower than the OLNE estimates (Fig.5(c)). In fact, those differences are consistent with changes in the regional OH estimates (Fig.5(d)). This finding implies that the changes in the NO$_x$ emissions estimates have affected the OH and HCHO abundances in these regions. Satellite data assimilation results reported by Miyazaki et al. (2017, 2020) indicate that NO$_x$ emissions in India have increased by 30% since 2008, whereas NO$_x$ emissions in China have declined since 2011 (Liu et al., 2016). Over E-China (Fig. 5(a &b)), the standard simulations reduce the absolute annual mean difference between OLNE and TROPOMI of $3 \times 10^{15}$ molecules cm$^{-2}$ to $1 \times 10^{15}$ molecules cm$^{-2}$, which is consistent with the lower NO$_x$ emissions in this region in the updated inventory (Fig . S8). Over India and SE-Asia, the standard OH concentrations are ~40% lower (Fig.5(d)) than the OLNE estimates, resulting in lower HCHO columns. The lower standard HCHO columns can be linked to the increasing NO$_x$ emissions in these regions (Fig.S8); however, the magnitude of the change in the NO$_x$ emissions for these regions in the updated inventory is likely overestimated.

In E-USA and W-USA (Table S3), the standard simulation reduces the MBE by 26% and 12%,
respectively. The reduction in MBE and RMSE values in Africa and South America is less than 10%.
Therefore, NOₓ emission uncertainties mainly affect the HCHO simulations in India and SE Asia.


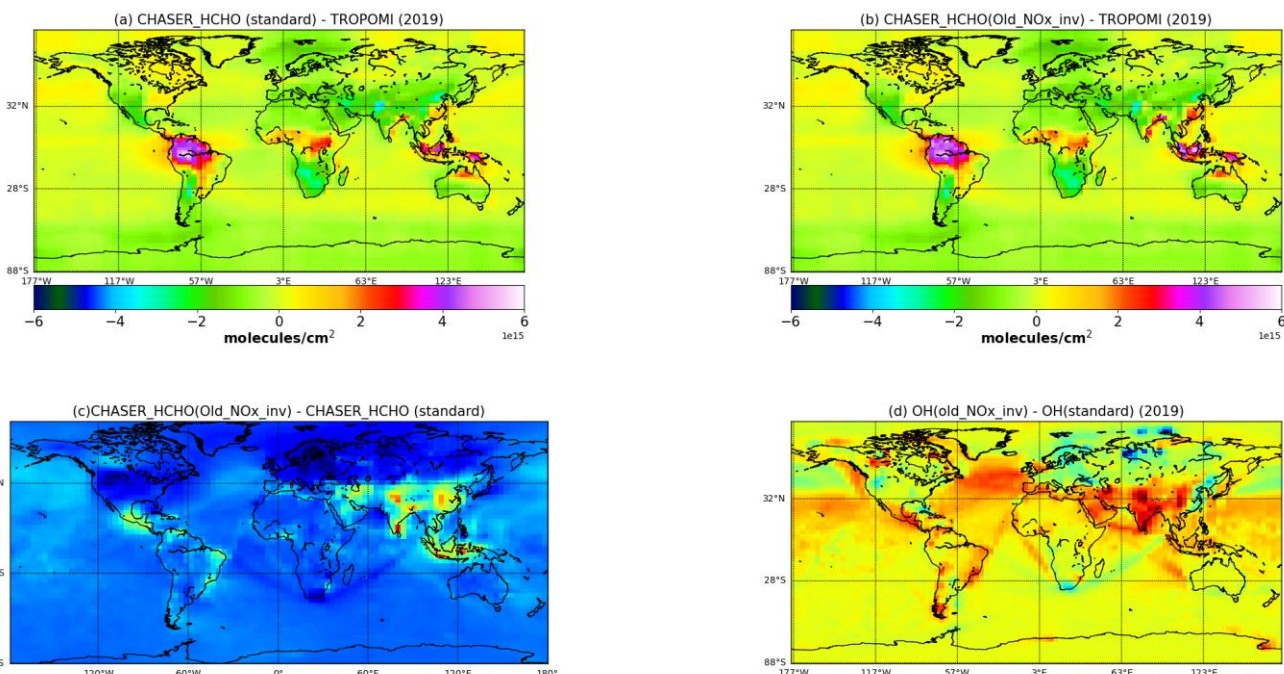

**Figure 5:** Annual mean HCHO columns ($\times 10^{16}$ molecules cm$^{-2}$) in 2019, obtained from the (a) standard and (b)
OLNE simulations. The HTAP-2008 NOx emission inventory was used instead of the HTAP-2018 inventory for
the OLNE simulations (Table 1). The remaining emission inventories are similar in both simulations. (c) Global
relative differences between the two HCHO simulations (OLNE–Standard). (d) Relative differences (global)
between two OH (OLNE–Standard) simulations. The standard and OLNE OH simulation settings are similar to the
description in Table 1. OH and HCHO simulations were obtained simultaneously.



## 3.5 Comparison with OMI HCHO Observations

TROPOMI was able to achieve improved precision of HCHO columns at shorter timescales (De Smedt et al., 2021). The effect of such features on the comparison results is evaluated in this section. The method of De Smedt et al. (2021) has been adopted to minimize the effect of different cloud retrieval algorithms used for OMI and TROPOMI retrievals. Figure S9 shows the global distribution mean HCHO columns obtained from TROPOMI and OMI retrievals and CHASER simulations in 2019 during the TROPOMI overpass time (13:30). Only the coincident dates among the three datasets are shown. Global and regional comparison statistics are presented in Table 6.

The spatial correlation between OMI and CHASER is 0.89 (Table 6) . OMI retrievals are positively biased by 7% compared to CHASER. A similar bias is also observed between TROPOMI and CHASER. Despite similar MBE values, TROPOMI reduces the global RMSE by 20%. Monthly MBE and RMSE values between OMI and CHASER are higher than those of TROPOMI and exhibit no seasonality (Table S3). The highest absolute differences between the model and OMI retrievals are observed in Amazonia in Brazil, C-Africa, and SE-Asia (Fig.S9). The magnitudes of differences between the model and observation in these regions are similar for both sensors. Despite the improved resolution, TROPOMI and OMI show equivalent biases in regions with high HCHO levels (De Smedt et al., 2021). A regional comparison among the three datasets is portrayed in Fig. 6. The red (TROPOMI–CHASER) and green (OMI–CHASER) numbers are the respective MBE values.

**Table 6.** Comparison of global mean HCHO columns between satellite observations (TROPOMI and OMI) and standard CHASER simulations. Units of MBE and RMSE are $\times 10^{16}$ molecules cm$^{-2}$. The $r$-value signifies the spatial correlation. The statistics are based on simulation and observations for 2019.

| Region | MBE1 (Standard– TROPOMI) | MBE2 (Standard– OMI) | RMSE1 (Standard– TROPOMI) | RMSE2 (Standard– OMI) | $r$-value (CHASER vs. TROPOMI) | $r$-value (CHASER vs. OMI) |
|---|---|---|---|---|---|---|

| | | | | | |
|---|---|---|---|---|---|
| Global | -0.23 | -0.24 | 0.77 | 0.99 | 0.93 | 0.89 |
| E-China | -0.84 | -2.54 | 1.40 | 3.03 | 0.56 | 0.17 |
| E-USA | 0.53 | -1.02 | 0.58 | 1.12 | 0.92 | 0.86 |
| W-USA | -0.72 | -2.09 | 0.80 | 2.17 | 0.83 | 0.64 |
| Europe | -0.78 | -1.31 | 0.92 | 1.60 | 0.77 | 0.67 |
| C-Africa | 1.19 | 0.94 | 1.57 | 1.28 | 0.93 | 0.93 |
| N-Africa | 1.46 | 1.42 | 1.61 | 1.59 | 0.81 | 0.79 |
| S-Africa | -0.99 | -2.59 | 1.32 | 2.75 | 0.86 | 0.84 |
| S-America | 2.99 | 2..02 | 3.41 | 2.61 | 0.47 | 0.56 |
| India | -1.05 | -1.19 | 1.57 | 2.66 | 0.85 | 0.66 |
| IGP | -1.22 | -2.85 | 1.69 | 3.19 | 0.91 | 0.84 |
| E-India | 0.26 | -0.05 | 1.22 | 1.34 | 0.82 | 0.76 |
| S-India | -0.59 | -0.16 | 0.69 | 0.41 | 0.96 | 0.97 |
| SE-Asia | -0.76 | -0.83 | 1.16 | 1.14 | 0.78 | 0.86 |




Over E-China (Fig.6(a)), the monthly mean TROPOMI columns are ~22% lower than those of OMI,
reducing the RMSE by 53%. The simulated spatial distribution shows better congruence with the new
observations. TROPOMI improved the summer model–satellite agreement considerably. The magnitude
of the seasonal modulation in the three datasets is 50%. Both sensors show that winter HCHO levels in
E-China are ~$8 \times 10^{15}$ molecules $cm^{-2}$.

Over E-USA (Fig.6(b)), the *r*-value between CHASER and OMI is 0.86. CHASER columns are
underestimated compared to OMI, with MBE and RMSE of $-1.0 \times 10^{15}$ and $1.1 \times 10^{15}$ molecules $cm^{-2}$.
TROPOMI reduced the model–satellite RMSE by 50% and improved the *r*-value by 6%. The most
significant improvements were observed during the summer and autumn.

Over the W-USA(Fig.6(c)), TROPOMI retrievals are 26% lower than OMI observations, reducing the
model–satellite RMSE by 63%. The spatial correlation between OMI and CHASER is moderate. The
simulated and TROPOMI wintertime columns are ~30% lower than OMI. However, the observed peak
in HCHO seasonality in July is consistent in the observational datasets.

OMI and TROPOMI HCHO observations over Europe(Fig.6(d))  are consistent. The seasonal cycle
amplitude inferred from both sensors is 60%. The simulated spatial distribution shows better agreement
with the TROPOMI observations, manifesting the effects of improved resolution.

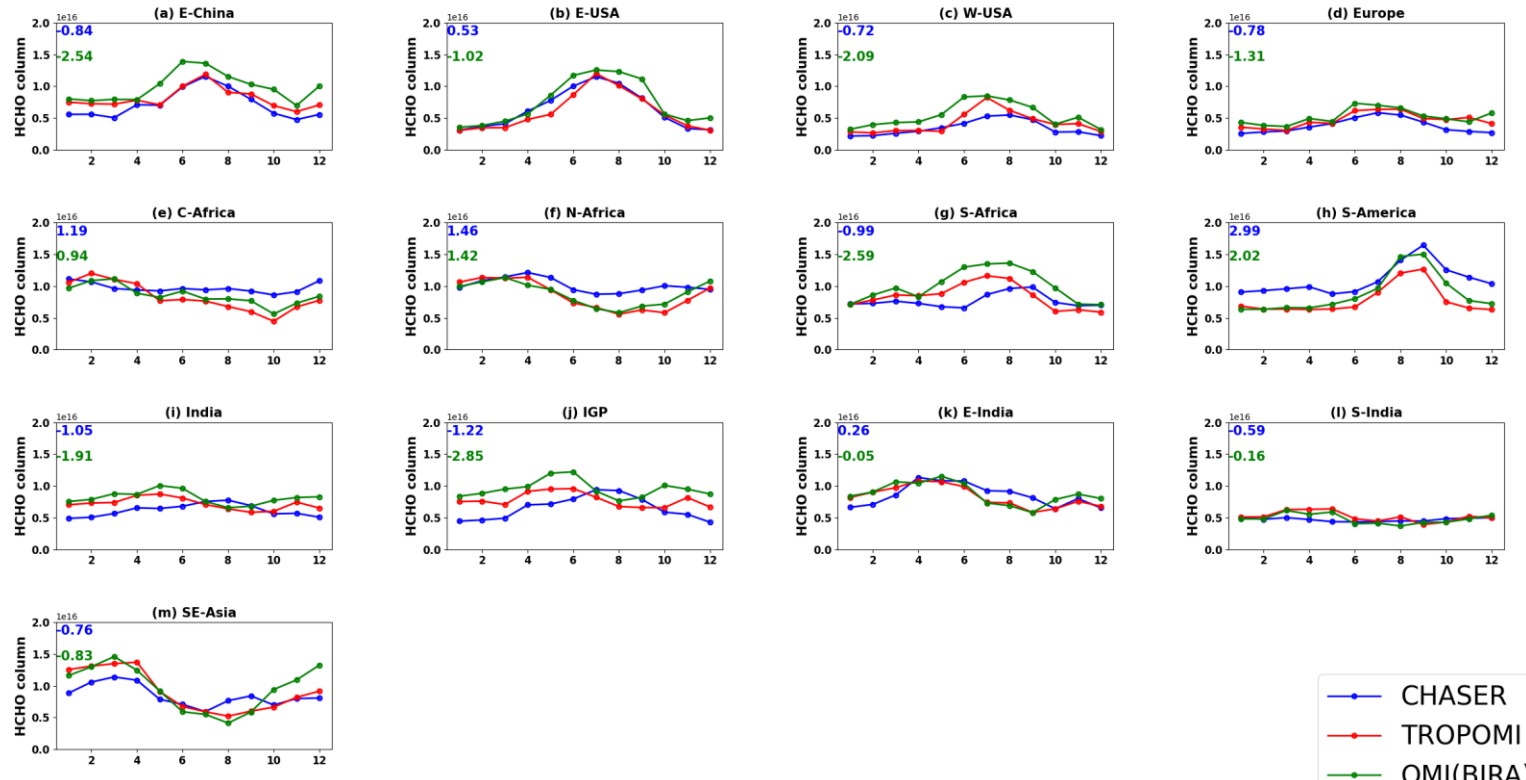

**Figure 6:** Seasonal variation of HCHO ($\times 10^{16}$ molecules cm$^{-2}$) inferred from TROPOMI (red curve) and OMI (orange curve) retrievals and standard CHASER (blue curves) simulations. The region definitions are shown in Fig. S2. The blue numbers signify the MBE between TROPOMI and CHASER, whereas the green numbers represent the MBE between CHASER and OMI. Coincident dates in 2019 among the datasets are used to calculate the monthly mean data.

Over C-Africa(Fig.6(e)), the RMSE value between CHASER and OMI is ~18% lower than that of TROPOMI. TROPOMI values are biased by 18% on the lower side compared to OMI.

Over N-Africa(Fig.6(f)), OMI retrievals are moderately correlated with CHASER. The amplitude of seasonal modulation inferred from CHASER, TROPOMI, and OMI are 48, 62, and 66%, respectively. The RMSE and MBE between OMI and CHASER are $1.41 \times 10^{15}$ and $1.59 \times 10^{15}$ molecules cm$^{-2}$, respectively. OMI retrievals are approximately 13% higher than TROPOMI. Simulated North African HCHO columns show better consistency with the observations during the biomass-burning season.

701

Over S-Africa(Fig.6(g)), OMI HCHO columns are biased respectively by 32 and 25% on the higher side compared to TROPOMI and CHASER. The simulated seasonal variabilities and spatial distribution of HCHO show more relevance to TROPOMI than to OMI.

705

Over S-America(Fig.6(h)), the simulated peak ($1.6 \times 10^{16}$ molecules cm$^{-2}$) in the HCHO seasonality shows strong congruence with the OMI observations. Despite such consistency, simulated values are higher than OMI retrievals, with MBE and RMSE of ~$2 \times 10^{15}$ molecules cm$^{-2}$. Observations and simulations show that the peak HCHO abundances can vary between $1.0 \times 10^{16} - 1.8 \times 10^{16}$ molecules cm$^{-2}$ in September. Although the $r$-value between OMI and CHASER is higher than that of TROPOMI, the model's capability to replicate the observed spatial distribution was limited. OMI HCHO columns are positively biased by 30% compared to TROPOMI, thereby reducing the model–satellite RMSE by 23%.

713

Over India(Fig.6(i)), CHASER HCHO columns are negatively biased by 23% compared to OMI observations. Although TROPOMI minimized the model–satellite bias, seasonal discrepancies between the model and observations prevail. Over the IGP region, OMI HCHO retrievals are biased by 24% and 36% respectively, respectively, on the higher side, compared to TROPOMI and CHASER. Both sensors captured a similar HCHO seasonality in the IGP, with a modulation of 49%. Although CHASER could not reproduce the seasonality, the simulated modulation is 48%. The bias between the model and observations (OMI and TROPOMI) is ~ 4% in E-India and S-India. Simulated HCHO spatial variation strongly correlates with the observation datasets ($r$-value of ~0.85). The amplitude of the seasonal modulation in E-India inferred from OMI is ~40%.

Over Southeast Asia(Fig.6(m)), CHASER columns are negatively biased by 19% compared to the OMI columns. Despite lower biases, both datasets have a similar model–satellite discrepancies during the dry season. A few reasons for the CHASER underestimation in SE Asia during the dry season have been discussed in section 3.2. In addition, assumptions and uncertainties in the retrieval could also potentially engender such model satellite discrepancy. Figure S10 compares CHASER and OMI SOA (González et al., 2016) products. The data selection criterion is similar to the description presented in Section 2. The

most relevant differences between the OMI BIRA and SAO products are related to the underlying CTMs that simulate the apriori profiles and the reference sector correction (Zhu et al., 2016). A comprehensive list of the differences between the two products is available from Zhu et al. (2016). The comparison statistics are given in Table S5. CHASER columns during the dry seasons in SE Asia show excellent agreement with the OMI SOA retrievals (Fig.S10(m)). OMI SOA values during the dry season are negatively biased by 7% compared to TROPOMI observations. The MBE between CHASER and SOA product is $0.04 \times 10^{15}$ molecules $cm^{-2}$. Based on the comparison with OMI SOA products, the model performance during the dry season can be considered excellent. The emission estimates for SE-Asia in CHASER can be regarded as reasonable, too.

Similarly, in E-China (Fig.S10(a)), the OMI SOA product reduces the bias between the model and observations by 11%. The simulated wintertime columns are consistent with the SOA estimates but underestimated compared to TROPOMI. The ANI estimates (Fig.4(a)) for this region are higher than the SOA product, manifesting that the anthropogenic emissions in CHASER for this region are rational. Therefore, uncertainties related to the retrieval procedure can also significantly affect the comparison results on a regional scale.

Comparison between CHASER and OMI BIRA HCHO products shows differences from the results of Hoque et al. (2022), where the simulation and observations for 2017 were used. The simulations in both studies are similar. However, the OMI data in the earlier study are systematically higher, mainly causing the statistically significant differences found between the study results. A detailed investigation of the reasons will be addressed in a separate work.

## 3.6 Validation using MAX-DOAS observations

### 3.6.1 Seasonal Variation

CHASER columns are compared with ground-based MAX-DOAS observations in Phimai, Chiba, and Kasuga in Fig. 7. Coincident TROPOMI observations over the sites are used for comparative discussion. The TROPOMI AK applied standard, and OLNE simulations are used. MAX-DOAS observations between 12:00 and 15:00 were averaged to estimate the monthly mean columns. Only the common dates among the three datasets were compared. De Smedt et al. (2021) compared the TROPOMI and A-SKY MAX-DOAS datasets in Phimai and Chiba. Because the model-ground-based comparison is the primary focus of this comparison effort, we do not consider the differences in the vertical sensitivity of TROPOMI and MAX-DOAS. Thus, the statistics will differ from De Smedt et al. (2021).

In Phimai, standard CHASER HCHO seasonality correlates strongly ($R$=0.71) with the MAX-DOAS observations; it is underestimated by 39%. However, the bias between the standard model estimates and TROPOMI observations is 4%. Despite a strong correlation, TROPOMI observations are negatively biased by 37% compared to the MAX-DOAS ($R$=0.84). Such underestimation might be related to the coarse binning of the satellite data. Using a finer bin, De Smedt (2021) reported a negative bias of 23% in Phimai.

Biomass burning-led enhancements during the dry season (January–April) are well reflected in the simulations. During the wet season, MAX-DOAS, TROPOMI, and standard CHASER HCHO columns are mostly lower than $1 \times 10^{16}$ molecules cm$^{-2}$. The simulated standard HCHO peak in March is consistent with the satellite observation, whereas MAX-DOAS observation shows a peak during February. During the dry seasons of 2015 and 2016, the HCHO peak was observed in March (e.g. Hoque et al., 2018). Consequently, such a shift in the HCHO peak might be related to fire numbers and fire radiative power changes (Hoque et al., 2022).

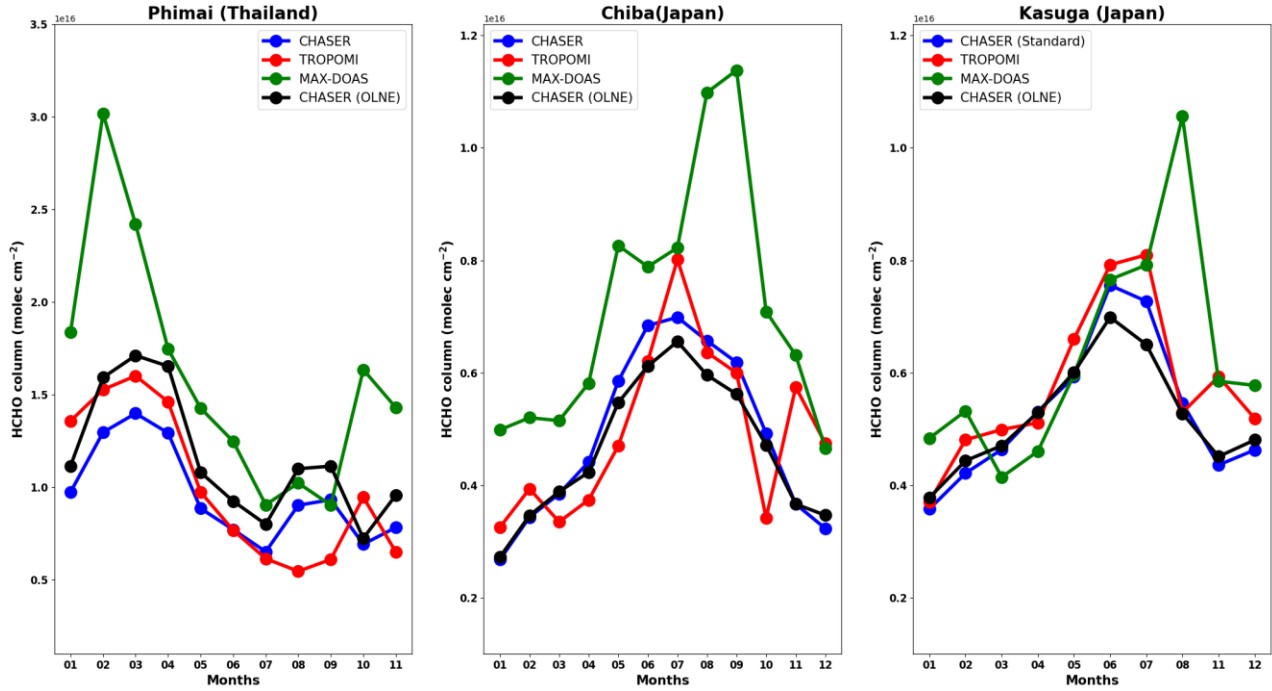

**Figure 7:** Seasonal variations in HCHO ($\times\ 10^{16}$ molecules cm$^{-2}$ cm$^{-2}$) columns inferred from satellite retrievals (red), model simulations (blue and black), and ground-based MAX-DOAS observations (green) in Phimai (Thailand), Chiba (Japan), and Kasuga (Japan). MAX-DOAS observations and CHASER simulations during 12:00–15:00 LT were selected for comparison. Common dates among the datasets are used to calculate the monthly mean statistics. The blue and black curves, respectively, signify the standard and OLNE simulations. TROPOMI AKs have been applied to both simulations. The simulation settings are provided in Table 1.



The bias between OLNE and MAX-DOAS observations is 27%. OLNE estimates agree better with the

TROPOMI observations during the dry season. However, the overall bias (13%) between the model and

satellite observations is higher in the case of OLNE simulations.


At Chiba, the simulated HCHO seasonality correlates strongly with the MAX-DOAS retrievals ($R$=0.81)
and is negatively biased by ~31%. The amplitudes of seasonality inferred from the simulations, MAX-
DOAS observations, and TROPOMI retrievals are, respectively, 59, 60, and 34%. The MAX-DOAS,
TROPOMI, and CHASER HCHO columns, respectively, reach peaks in September, July, and June.
Similar to Phimai, the HCHO peaks in satellite and ground-based observations differ. One reason might
be the differences in spatial representativity. TROPOMI data used for comparison are spatially averaged
over 200 km, centering on the Chiba site, whereas the spatial representativity of the MAX-DOAS is
approx−10 km. Moreover, MAX-DOAS observations are most sensitive to altitudes near the surface,
whereas satellite sensitivity decreases near the surface. Consequently, the air masses sampled by the
instruments at the same local time might differ, leading to inconsistent observation peaks.

At Kasuga, the simulated HCHO levels are strongly correlated with the TROPOMI observations ($R =$
0.75) and are negatively biased by 35%. Although the correlation between the model and MAX-DOAS
retrievals is moderate, the bias between CHASER and MAX_DOAS retrievals is 14%. Therefore,
CHASER shows better agreement with MAX-DOAS than with TROPOMI. MAX-DOAS observations
exhibit seasonality similar to that of Chiba, with a peak HCHO column during August. Similar to Chiba,
the satellite-observed and CHASER peaks are observed during July and June, respectively. Chiba and
Kasuga sites are located near the ocean and exhibit similar HCHO variability, which has been captured
well in the simulations.

Although the bias between OLNE and standard simulations for Chiba and Kasuga is ~4%, the absolute
difference is ~$1\times10^{15}$ molecules cm$^{-2}$. NO$_x$ emissions in Japan have not changed markedly since 2005
(Miyazaki et al., 2017). The differences between the simulations are observed during the summer when
isoprene emissions are expected to peak (Hoque et al., 2018a). Because the OH estimates over Japan are
similar for both simulations (Fig. 5(d)), the differences are likely related to the interaction between
isoprene and NO$_x$ inventories.

### 3.6.2 Diurnal and Daily Variations

Figure 8 compares the observed and simulated daily and diurnal variations in the surface HCHO vmr. The error bars represent the 1σ standard deviation of the observed mean values. The daily variation comparison entails only the standard simulations.

In Phimai, the daily datasets correlate well, with an *R*-value of 0.67. The slope of the fitted line is 0.35. The observed and simulated daily mean HCHO vmr is ~4 ppbv. CHASER daily mean values are negatively biased by 19% and 11%, respectively, during the dry and wet seasons. The standard diurnal variations at Phimai are also well correlated with the observations (*R*=0.64). The simulated values lie within the standard deviation of the observations. HCHO mixing ratios show a peak ( ~6 ppbv) at 8:00 LT in both datasets. Noontime (12:00 LT) vmr are approximately 4 ppbv, and hourly HCHO levels vary between 2 and 6 ppbv. The OLNE diurnal values are 20% higher than the standard values. However, the mean absolute difference between the two simulations is 1 ppbv.

The standard simulation reproduced the observed diurnal variations at Chiba, with a temporal correlation of 0.79, higher than at Phimai. Both simulations are biased by 10% on the lower side compared to the observations. No distinctive peak is observed in the diurnal variations. The increasing daytime HCHO levels in Chiba are well reflected in the model runs. The simulated daily mean values in Chiba are negatively biased by 18%, with a temporal correlation of 0.40. The slope of the fitted line to the daily mean concentrations is 0.27, lower than  at Phimai, suggesting a higher underestimation similar to the total columns (Fig. 7).

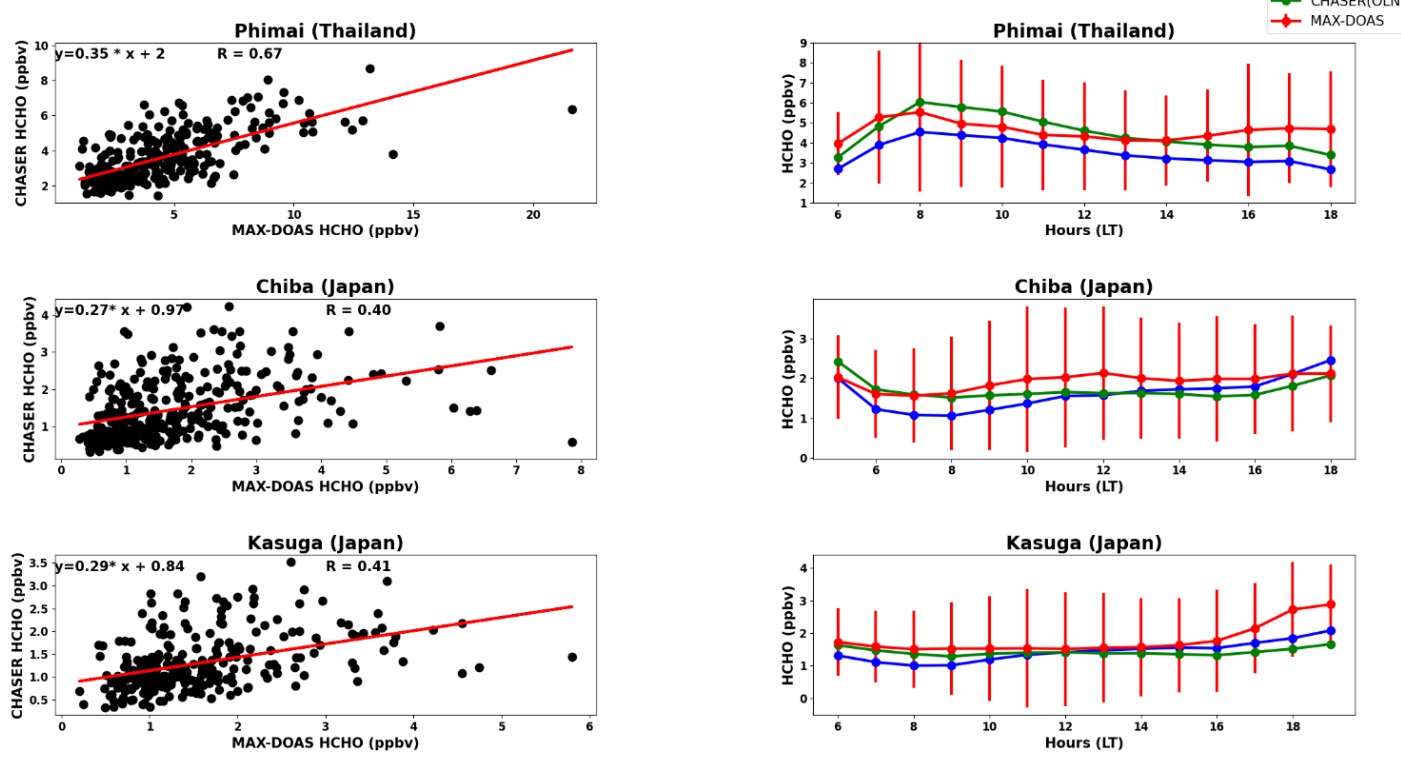

845

**Figure 8:** (left panel) Scatter plots show the correlation between the daily mean observed (MAX-DOAS) and simulated HCHO surface mixing ratios at the three sites. The standard simulations are used in the scatter plots. The linear fitted lines are shown in red. (right panel) Diurnal variations in the HCHO mixing ratios at the three sites are inferred from the MAX-DOAS observations and standard (blue) and OLNE (green) simulations. The error bars represent the 1-sigma standard deviation of the mean values estimated from the observations. Observations and simulations at the coincident date and time (local) are selected for comparison.



In Kasuga, modeled daily variations correlate moderately ($R$=0.41) with the observations. The effect of the $NO_x$ inventories on the simulated diurnal variations in Kasuga is not significant. The simulated daily mean values are negatively biased by 20%, and the slope of the fitting is 0.29. Although Chiba and Kasuga

are similar sites, their observed diurnal variations are slightly different. However, the simulated values in
both cases agree with the observed standard deviation.

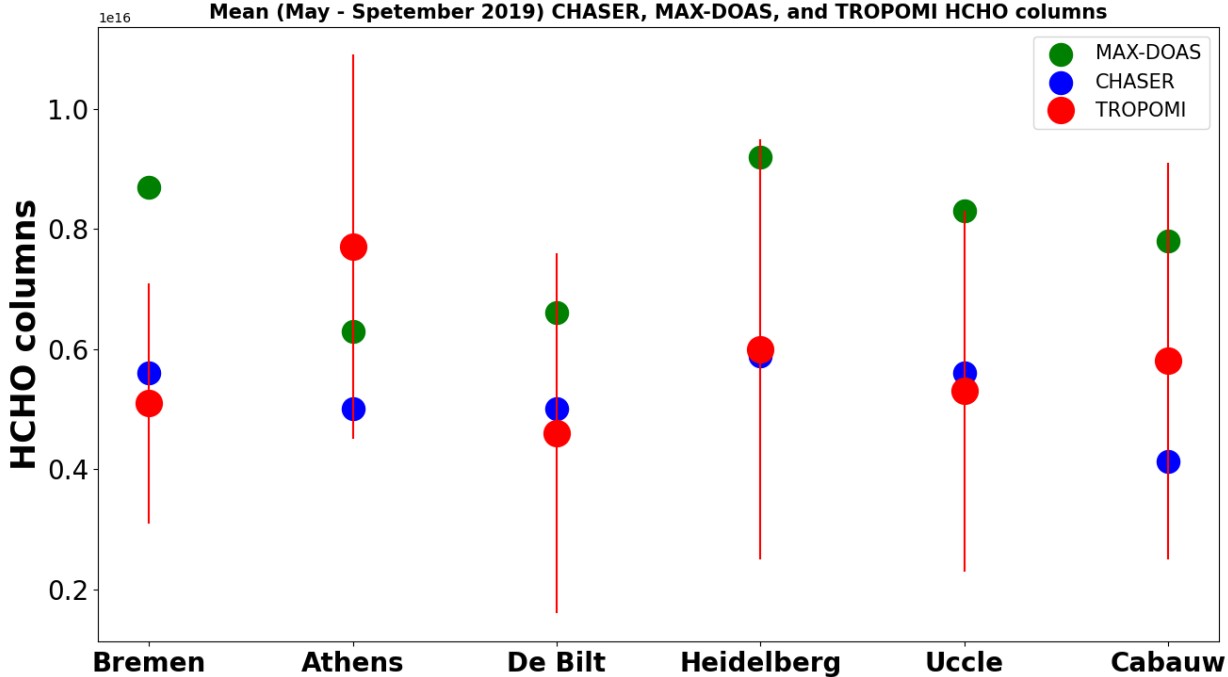

**Figure 9:** Scatter plot comparing CHASER (red), MAX-DOAS (green), and TROPOMI (red) HCHO columns
( $\times 10^{16}$ molecules cm$^{-2}$) at a few European sites. The MAX-DOAS observed values are taken from the work of
Oomen et al. (2024). These values represent the mean HCHO column from May to September in 2019. The
observations from 12:00 – 15:00 LT were used to calculate the mean values. Using a similar temporal filter, the
modeled mean values were calculated from the simulations for 2019. TROPOMI data for 2019 were filtered as
described in Section 2.2. The error bars signify the 1-sigma standard deviation of the TROPOMI mean HCHO
columns.



In addition, CHASER HCHO columns are also compared with MAX-DOAS observations reported in the
literature, shown in Fig.9. The observed values are obtained from Oomen et al. (2024). The observed

mean values represent the averages of MAX-DOAS observations between 12:00 and 15:00 LT from May to September 2019. A similar temporal filter was applied to the CHASER simulations for 2019. The coincident TROPOMI HCHO columns are also plotted. TROPOMI AKs are applied to the CHASER values. The error bars signify the 1-sigma standard deviation of the TROPOMI mean values.

Like the Asian sites, CHASER underestimates the HCHO columns at the European sites. All three datasets mostly agree within the 1-sigma variability range of the satellite observations. CHASER and TROPOMI HCHO columns are lower than the MAX-DOAS observations except in Athens. CHASER shows better agreement with the MAX-DOAS observations in Athens. De Smedt et al. (2021) reported the biases between TROPOMI and MAX-DOAS observations at these sites, estimated from a daily time scale. As the simulated HCHO magnitude is consistent with the TROPOMI values, biases between the CHASER and MAX-DOAS HCHO columns at these sites will likely be equivalent.

## 3.7 Comparison with ATom-4 flight observations

A comparison between simulated and observed HCHO and isoprene profiles along the ATom-4 flight path (Fig. S2) is depicted in Fig. 10 (a and c). Only the coincident dates have been included in the comparison.

The simulated HCHO and isoprene profiles agree well with the observations, with an *R*-value of 0.95. Above and below 4 km, CHASER HCHO profiles are positively biased by 29 and

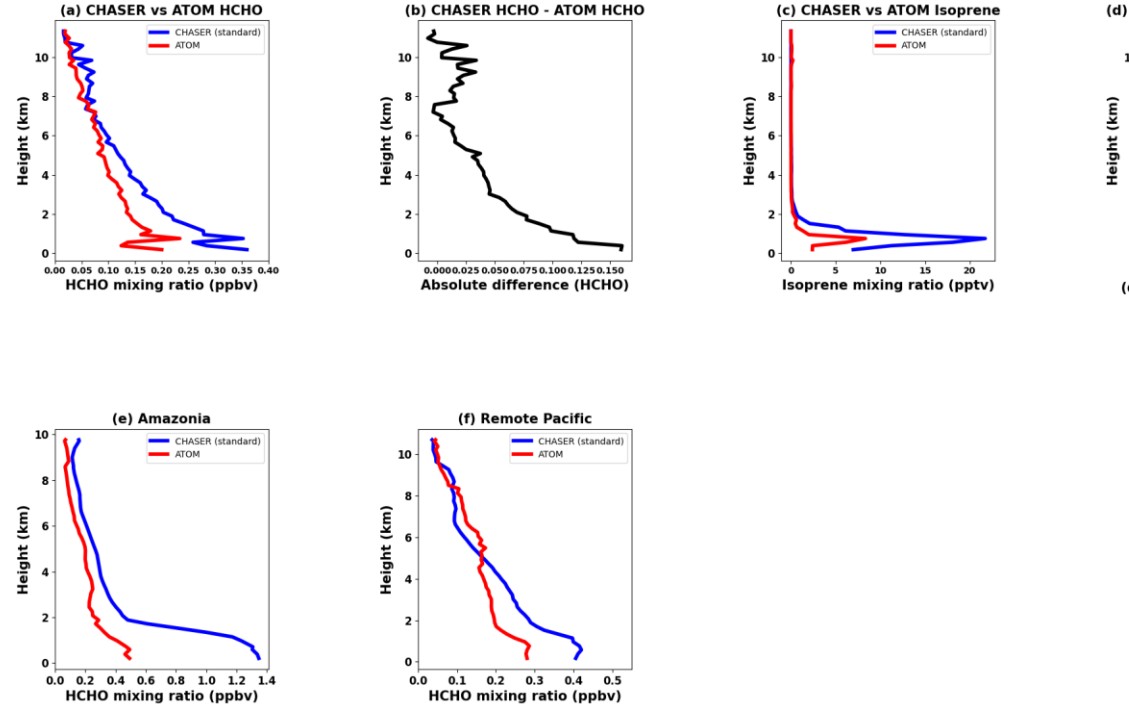

**Figure 10:** (top panel) Comparison between ATom observed (red) and CHASER simulated (blue) (a) HCHO, and (c) isoprene profiles along the ATom-4 flight path in 2018. The ATom-4 flight path is depicted in Fig.S2. Standard simulations are used for comparison. Simulations at the time of the ATom observations were selected. Both datasets were averaged within a 0.3 km bin. The relative differences between the observed and simulated (c) HCHO and (d) isoprene profiles are also shown. (bottom panel) Atom-4 observed, and CHASER simulated HCHO profiles over the (e) Amazonia and (f) the Remote Pacific region are compared. Amazonia (10º-40ºW,10ºS-10ºN) represents a densely vegetated region, whereas the remote Pacific region (160º-180ºW, 20ºS-20ºN) represents the background HCHO conditions. The units of the HCHO and isoprene mixing ratios are, respectively, ppbv and pptv.

62%, respectively, compared to ATom-4 HCHO levels. The absolute difference in the isoprene profiles around 1 km is 14 pptv, which strongly correlates with the difference in the HCHO profile below 2km. This finding signifies that overestimated CHASER isoprene mixing ratios induce a positive bias in the HCHO estimates. Despite non-significant isoprene mixing ratios at altitudes greater than 2 km, both

datasets show considerable HCHO levels above 2 km. Zhao et al. (2022) reported a similar finding and
attributed enhanced $CH_4$ oxidation to the HCHO mixing ratios above 2 km. At higher altitudes HCHO is
produced through the $CH_4$ oxidation (i.e., $CH_4$ + OH) initiated $CH_3O_2$ (methyl peroxy radical) + $CH_3O_2$
pathway. HCHO production through this pathway is considered in CHASER. Therefore, despite the
differences in the magnitude, CHASER has shown good skills in reproducing the VOC profiles.

The potential reason for the higher HCHO simulated values below 2 km could be CHASER's
overestimated HCHO mixing ratios over South America, mainly the Amazon (Fig 2(c). Figure 10(e and
f) depicts the observed and simulated HCHO profiles over the Amazon (10º-40ºW,10ºS-10ºN) and the
remote Pacific region (160º-180ºW, 20ºS-20ºN). The HCHO profiles over the remote Pacific region
represent the background HCHO mixing ratio. CHASER and ATom background HCHO mixing ratios
within the boundary layer are 0.4 and 0.3 ppbv, respectively. The mean relative differences between the
two datasets within the boundary layer over Amazonia and the remote Pacific region are ~60 and ~22%,
indicating that the uncertainty in the contributions from the isoprene emissions to the total HCHO
uncertainties is higher. Above 5 km, CHASER underestimates the background HCHO mixing ratios.
However, simulated and TROPOMI HCHO columns over the remote Pacific regions showed consistency
when gridded over a similar horizontal grid (Fig. 1). Consequently, differences in the horizontal resolution
can cause discrepancies between the simulations and ATom observations over the remote regions. Over
South America, the model overestimates the observed (TROPOMI and ATom) HCHO abundances
irrespective of the horizontal resolution. Therefore, the biogenic emission estimates for South America in
CHASER should be reviewed to reduce the model-observation biases.

**3.8 Contribution estimates**
The contributions of different VOC emission sources to the regional HCHO abundances are presented in
Fig. 11. The contribution estimates are presented in Table 8. A stacked-bar plot of the annual contributions
of the emission sources is portrayed in Fig. S11.

Over E-China (Fig.11(a)), biomass burning has a non-significant effect on the regional HCHO columns.
During summer, the biogenic and anthropogenic VOC emission contributions are 44% and 17%,
respectively. In contrast, anthropogenic and biogenic contributions to the regional HCHO level during
winter are 35% and 13%, respectively.

Non-significant biomass burning effects on the HCHO columns can be observed over E-USA (Fig.11(b)),
W-USA(Fig.11c)), and Europe(Fig.11(d)). Biogenic emissions contribute more than 20% (35% in E-
USA) in these regions. In these regions, annual anthropogenic contributions are higher than the biogenic
contribution. Although the simulated winter columns in these regions are consistent with TROPOMI (Fig.
2), the model values are lower during summer and autumn. Moreover, the sensitivity results show non-
significant biogenic contribution during winter and autumn, which likely reduces the annual biogenic
contribution estimates.

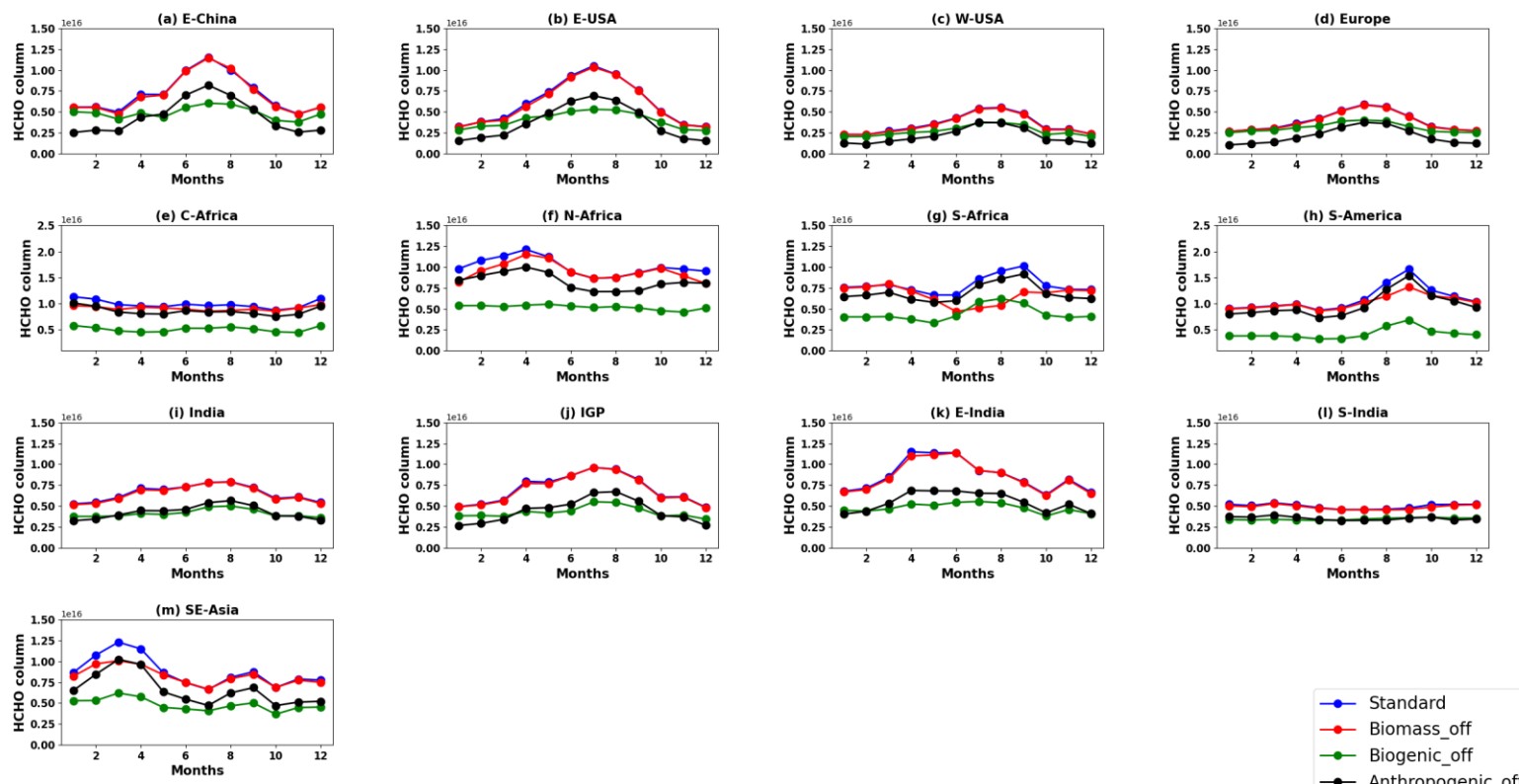


Figure 11: Seasonal variation of HCHO ($\times 10^{16}$ molecules cm$^{-2}$) inferred from different simulations. The settings of the standard simulation are presented in Table 1. The model estimates shown in red, green, and blue are simulated by switching off the biomass-burning, biogenic, and anthropogenic emissions. The satellite AKs have been applied to all the simulations. The coordinate bounds of the regions are similar to those in Fig. 2.

In C-Africa(Fig.11(e)), biogenic emissions (48%) are the most significant contributor, followed by anthropogenic emissions (13%). Although the biogenic emission contributions are equivalent in N-Africa(Fig.11(f); 48%) and S-Africa (Fig.11(b); 43%), the pyrogenic contributions are twice as high in the latter region. Consequently, despite similar HCHO abundances and modulation in these regions, the source contributions differ.

Table 8. Contributions (%) of different emission sources to HCHO abundances in selected regions. The respective emissions were switched off to estimate the contribution to the total HCHO abundances. The contributions have been calculated with respect to the standard simulations. The satellite AKs were applied to all simulations.

| Region | Biomass-burning contribution | Biogenic contribution | Anthropogenic contribution |
|--------|------------------------------|-----------------------|----------------------------|
| E-China | 1.4% | 32% | 37% |
| E-USA | 1.7% | 35% | 38% |
| W-USA | 1.8% | 23% | 39% |
| Europe | 1.2% | 20% | 45% |

| | | | |
|---|---|---|---|
| C-Africa | 8% | 48% | 13% |
| N-Africa | 6% | 48% | 17% |
| S-Africa | 15% | 43% | 12% |
| S-America | 7% | 61% | 10% |
| India | 1.4% | 37% | 34% |
| IGP | 1.1% | 39% | 37% |
| E-India | 1.5% | 44% | 36% |
| S-India | 2.1% | 30% | 29% |
| SE-Asia | 6% | 45% | 24% |

Biogenic emissions over South America(Fig.11(h)) contribute 61% to the regional HCHO abundances. The pyrogenic contribution during the biomass-burning period is 12%, whereas the annual contribution is 7%.

In SE-Asia(Fig.11(m)), annual anthropogenic contributions are ~20%. During the dry season, the anthropogenic, pyrogenic, and biogenic contributions are 7%, 12%, and 48%, respectively. Biogenic

production compromises 43% of the HCHO columns from July to December, whereas anthropogenic
emissions account for 9%.

In India(Fig.11(i)), annual pyrogenic emissions contribute ~2% to the HCHO levels. A similar source
contribution to the HCHO levels in IGP(Fig.11(j)) is also observed. The model's capability to reproduce
the observed HCHO seasonality in India and the IGP region was limited. Consequently, robust source
contribution estimates for these regions cannot be derived from the current analysis.
Over E-India (Fig.11(k)), 44% of the HCHO levels originate from biogenic sources, followed by
anthropogenic VOC emissions (36%). Similar source contributions of biogenic (30%) and anthropogenic
(29%) emissions are observed in S-India(Fig.11(l)). Over both regions, the pyrogenic source contribution
is ~2%.

## 985 3.9 Uncertainties in the chemical mechanism

Uncertainties in the chemical mechanisms affect the HCHO simulations. Representation of isoprene
chemistry can vary among the gas-phase chemistry mechanisms used in the CTMs. The most commonly
used isoprene schemes underestimate observed HCHO by at least 15% (Marvin et al., 2017). Such
underestimations are also strongly linked with the errors in the $NO_x$ emission inventories (Anderson et
al., 2017). In addition, potential errors in the acetaldehyde emission and chemistry can also lead to
underestimated HCHO vmr up to 75 pptv in the lower troposphere (Anderson et al., 2017).

## 993 4 Conclusions

CHASER simulated global HCHO spatiotemporal distributions at a horizontal resolution of 2.8° ×
2.8°were evaluated against multi-platform observations. First, two years of simulation results (2019–
2020) were compared with the latest HCHO satellite observations from TROPOMI. The model-satellite
agreement was excellent, with a global $r$-value of 0.93 and RMSE of $0.75 \times 10^{15}$ molecules cm$^{-2}$. The
model showed good capabilities for reproducing the HCHO columns in hotspot and background regions.
CHASER HCHO columns over large forested areas showed good consistency with the observations,
demonstrating that the biogenic emission estimates in the model are reasonable. Simulated HCHO
seasonality in a few selected regions was consistent with the observations. The model was able to

reproduce the observed wintertime HCHO columns in E-USA, W-USA, and Europe, in addition to summer peaks. Disagreement between TROPOMI and CHASER was observed primarily in India, China, Amazonia, and SE Asia. Uncertainties in background HCHO columns, anthropogenic VOC emission inventories, chemical mechanisms adopted in the model, and retrieval algorithms were the potential contributors to these discrepancies. However, such uncertainties did not affect the model–satellite agreement in Africa and South America. Comparison among OMI, TROPOMI, and CHASER HCHO columns demonstrated that TROPOMI's improved spatial resolution effect was limited globally. However, in most regions, simulated HCHO seasonality showed better agreement with TROPOMI than with OMI, reducing the RMSE by up to 63%. TROPOMI retrievals were, on average, 30% lower than those of OMI.

Second, CHASER simulations were compared with two-year MAX-DOAS observations of HCHO at Phimai, Chiba, and Kasuga. Daily CHASER HCHO mixing ratios showed consistency with the observations at the three sites, with $R$-values of 0.39–0.67. The slopes of linear fitting were lower for Chiba (0.29) and Kasuga (0.29) than for Phimai (0.37), implying lower model underestimation at the latter site. The diurnal variations at the sites were consistent with the observations. The change in the $NO_x$ emission inventories did not affect the simulated diurnal variations.

Third, simulated HCHO and isoprene profiles for 2018 were compared with ATom-4 flight observations. Despite consistent profile shapes, the model overestimated VOC mixing ratios mainly within the PBL. Uncertainties related to VOC emission inventories, background HCHO levels, and model resolution were potential reasons for the model–flight discrepancies.

Lastly, sensitivity studies were conducted to estimate the contributions of the different emissions sources to the total HCHO columns in different regions. Biogenic emissions were the most significant contributor in most of the regions. In a few cases, biogenic and anthropogenic emission contributions were equivalent. In some regions, only summertime biogenic estimates were found to be reasonable.

**Code availability**: The CHASER source code needed to reproduce the simulations in this work is available from the repository at https://zenodo.org/records/10892945 (Sudo et al., 2024).

**Data availability:** The processed model output and observational datasets needed to reproduce the results are available from the repository at https://zenodo.org/records/10052384 (Hoque et al., 2024). The MAX-DOAS profile and column data provided by Dr. Hitoshi Irie can be accessed from the repository(i.e., Hoque et al., 2024). TROPOMI (https://scihub.copernicus.eu/dhus/#/home, last access: 01 July 2023; De Smedt et al., 2021), OMI BIRA product, (https://www.temis.nl/qa4ecv/hcho/hcho_omi.php, last access: 01 July 2023; De Smedt et al., 2021) and ATom(https://daac.ornl.gov/ATOM/guides/ATom_nav.html, last access: 01 July 2023; Wofsy et al., 2018) data were obtained from the respective websites.

**Author contributions**: HMSH conceptualized the study, conducted the model simulations, analyzed the datasets, and drafted the manuscript. YH helped with the data processing. HI developed the JM2 code and maintained the A-SKY network. KS developed the CHASER model and supervised the study. MFK extended his expertise to explain the results. All the authors commented and provided feedback on the final results and manuscript.

**Conflict of Interest:** The authors declare that they have no conflict of interest

**Acknowledgments:** We are grateful to the TROPOMI, OMI, and ATom scientific teams for making the respective observational datasets available for public usage. The CHASER model simulations are partly performed with the supercomputer (NEC SX-Aurora TSUBASA) at the National Institute for Environmental Studies (NIES), Tsukuba, Japan. The corresponding author acknowledges the valuable advice of Dr. Kazuyakai Miyazaki (Jet Propulsion Lab, NASA) and Dr. Takashi Sekiya (JAMSTEC, Japan). The research has been supported by the Ministry of the Environment, Government of Japan (Global Environmental Research Fund (grant nos. S-12 and S-20)), the Japan Society for the Promotion of Science (KAKENHI (grant nos. JP20H04320, JP19H05669, JP19HO4235, JP23H04971, JP21K12227, JP22H03727, and JP22H05004)), the Environment Research and Technology Development Fund (JPMEERF20215005) of the Environmental Restoration and Conservation Agency of Japan, and the JAXA 3rd research announcement on the Earth observations (grant number 19RT000351).

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
