# Peer review of "Evaluating CHASERV4.0 global formaldehyde (HCHO) simulations"

_EGUsphere, 2024_

## Referee Comment (RC1)

**Review**

Hoque et al., using the observation results of Tropospheric Ozone Monitoring Experiment (TROPOMI), Atmospheric Tomography Mission (ATom) and Multi-axis Differential Optical Absorption Spectroscopy (MAX-DOAS), evaluated the HCHO simulation obtained from the global chemical transport model CHASER with a horizontal resolution of 2.8 × 2.8 in the past two years. The structure of the manuscript is reasonable and the amount of data processed is relatively large. And the content of this manuscript conforms this magazine. I recommend this paper for publication after minor revisions.

**Major comments**

1. The abstract describes the evaluation of HCHO simulation obtained from global chemical transport model CHASER with horizontal resolution of 2.8 × 2.8 in the past two years. The abstract is a highly concise summary of the full manuscript. The abstract of the manuscript should indicate which two years of data were evaluated. Please add relevant content in the abstract.

2. In the manuscript abstract, the variation of the simulation time (daily and diurnal) of formaldehyde mixture ratio has a good correlation with the MAX-DOAS observation results. However, the correlation R=0.41 or R=0.40 in the manuscript does not show that there is a good correlation between the two results.This part is not properly described by the author, please make adjustments.

3. The observed value of MAX-DOAS is generally column concentration, while the HCHO value observed by MAX-DOAS in this manuscript is ppbv. How did the author make the conversion? Please add the description of relevant content.

4. In the manuscript content, it is described that the MAX-DOAS observations between 12:00 and 15:00 are selected for average, and then in Figure 8, the author expresses the meaning of choosing the observations between 12:00 and 14:00. Please check it carefully.

**Detailed comments**

1. Line 296:The HCHO columns' peaks are compatible with the peak in isoprene concentrations (Fig. S3), manifesting a strong biogenic contribution during summer.In Fig.S3, the legend is not complete, please check it carefully.

2. Line 314:Fig.S2 isATom-4 flight track, which cannot be seen the peaks in the HCHO variability coincide with the isoprene peak in these regions

3. Line 387:Both datasets show enhanced HCHO levels during spring., consistent with high isoprene concentrations (Fig.) The manuscript does not indicate which drawing it is, please check it carefully.

---

## Referee Comment (RC4)

**Review of "Evaluating CHASER V4.0 global formaldehyde (HCHO) simulations using satellite, aircraft, and ground-based remote sensing observations" by Hoque et al.**

This paper presents the evaluation of HCHO columns from the CHASER model against the TROPOMI, OMI, ground-based MAX-DOAS observations, and the CHASER HCHO vertical profiles against the Atom-4 flight dataset. The authors compare the modelled regional HCHO columns with the TROPOMI and the OMI HCHO columns and analysed the model-observation differences comprehensively. The authors also compare the modelled HCHO columns with the MAX-DOAS columns at three locations in Thailand and Japan respectively. The modelled HCHO profile and the profile from the Atom-4 flights are compared for Amazonia and for the Remote Pacific region, respectively. The authors have also performed sensitivity simulations to assess the impact of anthropogenic, biogenic, and biomass burning VOC emissions, as well as NOx emissions on modelled HCHO. However, I find that one limitation is the lack of discussions on the important role of chemical mechanisms in simulating HCHO in the models, despite that the authors did mention this in the conclusion. There are some previous studies that the authors could cite which addressed inter-model differences in modelled HCHO (see below suggestions).

Overall, the analysis is thorough and robust. The paper is generally well-written, and the materials are well organised, and is within the scope of GMD. However, the presentation of the paper can be improved. I encourage the authors to make a thoroughly revision of the manuscript.

Below are two relevant papers on model differences in modelling HCHO (and CO):

*Anderson, D. C., Nicely, J.M., Wolfe, G. M., Hanisco, T. F., Salawitch, R. J., Canty, T. P., ... Zeng, G. (2017). Formaldehyde in the tropical western Pacific: Chemical sources and sinks, convective transport, and representation in CAM-Chem and the CCMI models. Journal of Geophysical Research: Atmospheres, 122. https://doi.org/10.1002/2016JD026121 (Figure 13)*

*Zeng, G., Williams, J. E., Fisher, J. A., Emmons, L. K., Jones, N. B., Morgenstern, O., Robinson, J., Smale, D., Paton-Walsh, C., and Griffith, D. W. T.: Multi-model simulation of CO and HCHO in the Southern Hemisphere: comparison with observations and impact of biogenic emissions, Atmos. Chem. Phys., 15, 7217–7245, https://doi.org/10.5194/acp-15-7217-2015, 2015. (Figure 15 and Table 4)*

My specific comments are listed below.

Abstract

It feels that the abstract is overly concise and does not reflect fully what are presented in the paper.

L19-20: Please state which comparison this is for, i.e., TROPOMI.

L30: It is the comparison between the CHASER and MAX-DOAS HCHO columns, not mixing ratio. Please also state the disagreement, i.e., CHASER underestimates the HCHO peak in comparison with the MAX-DOAS data at all three locations. You speculate that the model data

averaged over a large area might not be able to capture the observed peak at these locations. A mention of this would be useful in the abstract.

Introduction

L82: How do you evaluate OH?

Model, observations, and methods

L93: Is there a reference for this?

L99-101: A list of the reactions in a table (in supplementary) could be considered if they have not been published before.

L110-123: It will be helpful to tabulate these emissions.

L117: Do you calculate lightning NOx emissions online or prescribe them?

L124-125: Are there OH observations from OMI and Atom? Please provide details.

Table 1: ANI and OLNE appear first time in Table 1. Please define these simulations in the text.

L137: What are the TROPOMI grids?

L139: Do you mean that the TROPOMI data are interpolated onto the CHASER horizontal grid?

L174: Should be "2.3 OMI"

Results and discussion

L231: This section is essentially the comparison of CHASER HCHO with TROPOMI. Maybe "TROPOMI" should be reflected in the section title?

L235-239: I am not sure how meaningful these statistics are in terms of the global means as the global HCHO distribution is so inhomogeneous.

L243-245: Would it more suitable to note this in the MAX-DOAS comparison section?

Table 2: These numbers don't have to be in a table. You could include them in the Figure 1 caption. Is the correlation coefficient spatial or temporal?

Figure2: The panels can be larger. Mark the position of the MBE numbers in the panels consistently. Add identifiers to the sub-figure, e.g., (a), (b), ... for each region. Then refer to Figure2(a), Figure 2(b), etc., when you discuss them in the following subsections.

L293-: Please refer to the figure(s) and table(s) that your discussions are based on at the beginning of each subsection. Same as the following subsections of (b), (c), etc.

L300: Do you mean direct HCHO emissions or indirect (degradation of VOCs) HCHO emissions? Please clarify.

L319: Can you speculate what drives these model-satellite discrepancies in the Europe and W-US in summer and autumn?

L328-329:  Please refer to the figure you are referring to. Please also note that the C-Africa off-peak HCHO is overestimated by CHASER compared to TROPOMI (Figures 1 and 2).

L333: Figure S4: There is no black curve in Figure S4. Please revise this figure or the caption.

L338: Please mention the figure you refer to for these discussions.

L339: Missing ")" in "(De Smedt et al., 2008". Again, please mention the figure here you are referring to over the next few lines.

L340-341: "*The lower CHASER columns in Southern Africa are likely attributable to underestimated pyrogenic emissions.*" - Can you confirm this from the following sensitivity simulations?

L355-356: Why particularly mention the biomass burning in N Africa here?

L358-359: Could the chemical mechanism in the model be at play?

L370: Refer to relevant figure(s) and table(s) earlier in your discussion of the results. Do you have an estimate how the biogenic and biomass burning emissions in India compared to other regions? Are there any specific meteorological conditions in this region that lead to low HCHO and the lack of seasonality?

L387: The figure number is missing here.

L422-423: I am not sure what you try to convey here?

Figure 5: Could you increase the size of the panels in this figure?

L552: It is important to summarise the NOx emissions in the two inventories you used. What are the differences in NOx emissions between these two inventories? It will help to understand the impact of NOx on HCHO and OH.

L553: you need to define the OLNE simulations before referring to it.

L565-566: Which figure that you are referring to here?

Figure 6: It will be helpful to understand this figure if the differences in NOx emissions are displayed or mentioned.

L595-: This section should be condensed where appropriate. You have compared CHASER and TROPOMI HCHO columns in detail already, so the focus here should be on what those most significant differences between OMI and TROPOMI HCHO are and how they compare with the CHASER HCHO.

L612: Referring to Figure 7 at the beginning of this paragraph.

L680: Which "observation" do you refer here?

L688-689, L703: what are differences between OMI SOA and OMI BIRA HCHO products? A brief introduction will be helpful.

L762-764: Is this coincidental?

L808: "*In Kasuga, modelled diurnal variations correlate strongly (R=0.85) with the observations*". But in Figure 9, the R value is 0.41, not 0.85. Please check.

L837-839: Could you elaborate a bit more on this mechanism?

L844: Please check the coordinates for Amazon. You could draw two boxes on the map (Fig. S2) to represent the two studied regions.

L872-873: Have you already defined these sensitivity simulations?

L932-934: Does the model's course resolution play a role in this case?

L944: The last half sentence doesn't read well; do you mean the model underestimates the biogenic contributions?

---

## Author Comment (AC2)

**Responses to the reviewer 1**

The review comments and our responses are colored in blue and black texts, respectively. The changes in the manuscript corresponding to the comments are highlighted in yellow. The line and figure numbers refer to the revised manuscript.

Hoque et al., using the observation results of Tropospheric Ozone Monitoring Experiment (TROPOMI), Atmospheric Tomography Mission (ATom) and Multi-axis Differential Optical Absorption Spectroscopy (MAX-DOAS), evaluated the HCHO simulation obtained from the global chemical transport model CHASER with a horizontal resolution of $2.8 \times 2.8$ in the past two years. The structure of the manuscript is reasonable and the amount of data processed is relatively large. And the content of this manuscript conforms this magazine. I recommend this paper for publication after minor revisions.

We thank the reviewer for the insightful comments, which helped improve the manuscript's quality.

**Major comments**

1. The abstract describes the evaluation of HCHO simulation obtained from global chemical transport model CHASER with horizontal resolution of $2.8 \times 2.8$ in the past two years. The abstract is a highly concise summary of the entire manuscript. The abstract of the manuscript should indicate which two years of data were evaluated. Please add relevant content in the abstract.

**Response:** Simulations for 2019 and 2020 have been validated with satellite observations in the same years. We have included this information in the abstract.

L15-19 Two years (2019 -2020) of HCHO simulations obtained from the global chemistry transport model CHASER at a horizontal resolution of $2.8° \times 2.8°$ have been evaluated using observations from the Tropospheric Ozone Monitoring Experiment (TROPOMI), Atmospheric Tomography Mission (ATom), and multi-axis differential optical absorption spectroscopy (MAX-DOAS) observations.

2. In the manuscript abstract, the variation of the simulation time (daily and diurnal) of the formaldehyde mixture ratio has a good correlation with the MAX-DOAS observation results. However, the correlation R=0.41 or R=0.40 in the manuscript does not show that there is a

good correlation between the two results.This part is not properly described by the author, please make adjustments.

**Response:** We have revised the lines, and the following lines have been included.

L31-34: The simulated seasonal variations in the HCHO mixing ratio showed good agreement (R > 0.70) with the MAX-DOAS observations and agreed within the 1-sigma standard deviation of the observed values. However, the temporal correlation (R~0.40) was moderate on the daily scale.

3. The observed value of MAX-DOAS is generally column concentration, while the HCHO value observed by MAX-DOAS in this manuscript is ppbv. How did the author make the conversion? Please add the description of relevant content.

**Response:** The volume mixing ratio is calculated using the partial VCD, U.S. standard temperature, and pressure. Temperature and pressure data were scaled to the respective site's surface measurements. The following texts have been included in the revised manuscript.

L259-261: The partial VCD values are converted to the volume mixing ratio  (vmr) by scaling the  U.S. standard atmosphere temperature and pressure data to the respective site surface measurements.

4. In the manuscript content, it is described that the MAX-DOAS observations between 12:00 and 15:00 are selected for average, and then in Figure 8, the author expresses the meaning of choosing the observations between 12:00 and 14:00. Please check it carefully.

**Response**: We thank the reviewer for identifying the mistake. The revised version is as follows:

**Figure 8:** Seasonal variations in HCHO ($\times\ 10^{16}$ molecules cm$^{-2}$ cm$^{-2}$) columns inferred from satellite retrievals (red), model simulations (blue and black), and ground-based MAX-DOAS observations (green) in Phimai (Thailand), Chiba (Japan), and Kasuga (Japan). MAX-DOAS observations and CHASER simulations during 12:00–15:00 LT were selected for comparison. Common dates among the datasets are used to calculate the monthly mean statistics. The blue and black curves, respectively, signify the standard and OLNE simulations. TROPOMI AKs have been applied to both simulations. The simulation settings are provided in Table 1.

**Detailed comments**

1. Line 296: The HCHO columns' peaks are compatible with the peak in isoprene concentrations (Fig. S3), manifesting a strong biogenic contribution during summer.In Fig.S3, the legend is not complete, please check it carefully.

**Response:** We have revised the Fig S3 which is Fig.S4 in the revised manuscript.

2. Line 314: Fig.S2 isATom-4 flight track, which cannot be seen the peaks in the HCHO variability coincide with the isoprene peak in these regions

**Response:** We have revised the text.

3. Line 387: Both datasets show enhanced HCHO levels during spring., consistent with high isoprene concentrations (Fig.) The manuscript does not indicate which drawing it is, please check it carefully

**Response:** We have added the missing figure, which is FigS6 in the revised manuscript.
* * *
**Responses to the reviewer 2**

The review comments and our responses are colored in blue and black texts, respectively. The changes in the manuscript corresponding to the comments are highlighted in yellow. The line and figure numbers refer to the revised manuscript.

I trust this meassage finds you in good health. I am writing to offer my review of your article titled "Evaluating CHASER V4.0 global formaldehyde (HCHO) simulations using satellite, aircraft, and ground-based remote sensing observations" After a thorough evaluation of your research, I would like to extend my appreciation for the valuable contributions your work brings to the field.There are some areas where minor enhancements could be made to elevate the clarity and coherence of the text. I would like to highlight these areas for your consideration:

We thank the reviewer for the insightful comments

Line 70: Any result or outcome that reflects that "good agreement"?

**Response:** We have added the following information in response to this comment.

L75– 77 Ryan et al. (2021) and Chan et al. (2020) reported good agreement (temporal correlation, $R > 0.70$) between TROPOMI and MAX-DOAS in Melbourne and Munich

Line 143: Reprocessed and Offline TROPOMI products are different datasets. Please provide more detail about how and when each one is used.

**Response:** We thank the reviewer for identifying this issue. We have used the offline product only. The revised version is as follows:

L163-164: The TROPOMI operational L2 offline (OFFL) HCHO vertical column density (VCD) (ver. 1.1.5.7) data from 2019 to 2020 have been used for this study.

174: Typo: "2.2" where it should read "2.3".

**Response:** 2.2 has been replaced with 2.3

190: 2018 is out of the original study period (2019-2020). Please clarify why this year was chosen.

**Response:** The Atom campaign spans from 2016 to 2018. Thus, we used the 2018 simulations for comparison. This has been updated in the revised abstract.

194-196: Typo: "TOGO" where it should read "TOGA" (Trace Organic Gas Analyzer).

**Response:** We have corrected the typo error.

201: Why were those specific locations chosen? Please provide more detail about that decision.

**Response:** The sites were selected because continuous observations from 2019 to 2020 were available for these sites. We have added the following line in the revised manuscript.

L238-239 The sites were selected because continuous measurements from 2019 to 2020 were available for these sites

208: Typo: "Kasugai" where it should read "Kasuga".

**Response.** Kasugai has been changed to Kasuga

257: Figure 1 is a valuable result and should appear in a larger size. With the current layout, the visualization is very difficult.

**Response:** We thank the reviewer for the comment. Figure 1 has been revised.

279: Figure 2 and similar ones: The temporal axis should specifically detail the study period, indicating whether it represents an average of both study years (2019-2020).

**Response**. We thank the reviewer for the comment. We have mentioned about the study period in each figure and Table caption. We have also mentioned it in the text. We adopted this method, because we found that mentioning the time period for every subplot reduced the clarity of the figures.

467: Typo: "Fig. ", without a number.

**Response:** The Figure number is S6 which has been added in the revised manuscript.

489: Please place the MBE values always in the same position in the graphs.

**Response:** The position of the MBE values has been fixed in the revised figures.

725: THat is still a high negative bias. Are there any further possible reasons for this result?

**Response:** The differences in the ground-based and satellite sensor sensitivity can also be a potential reason for the significant biases. This has been discussed in the following lines:

L799-801: Moreover, MAX-DOAS observations are most sensitive to altitudes near the surface, whereas satellite sensitivity decreases near the surface. Consequently, the air masses

sampled by the instruments at the same local time might be different, leading to inconsistent observation peaks.

756: TROPOMI spatial resolution is 3.5 x 5.5 km$^2$. Would there be any chance to perform this comparison at a higher resolution instead of 200 km spatial averaging?

**Response**: Yes, the comparison can be performed at a higher resolution. CHASER can run on a horizontal resolution of 2.8º, 1.4º, and 0.56º. 2.8 is the most commonly used horizontal resolution for CHASER studies (global and data assimilation), so it has been chosen for evaluation. The supplementary information includes a comparison (FigS1 and Table S1) between simulations performed on two horizontal resolutions (2.8 º and 1.4 º).

The difference between the TROPOMI and CHASER horizontal resolution will lead to representative errors. However, the random horizontal representative errors are mostly limited to 5-10% and are included in the individual retrieval error (Boersma et al., 2015). The most relevant representative error is associated with the vertical sensitivity of the satellite sensor. Our simulations have accounted for the averaging kernel information needed to address this issue. A detailed assessment of the horizontal resolution effect on the comparison results is out of the scope of the current study. We will address this issue separately. However, we have added the following text in the revised manuscript.

L196-205 TROPOMI observations are averaged spatially and temporally to the CHASER grid (T42) daily, leading to horizontal representativeness errors. However, the random horizontal representativeness errors are in the order of 5-10%, which is lower than the individual retrieval error of the satellite observations (Boersma et al., 2015). If the model horizontal resolution is increased by 50% (i.e., simulated at a horizontal resolution of 1.4º × 1.4º), the change in HCHO abundances is less than 6% (Fig S1 and Table S1 in supplementary information). The vertical sensitivity of the satellite retrievals is the most relevant source of representativeness error (Boersma et al., 2015). The current study utilizes the TROPOMI AK information to minimize the representativeness error. Therefore, the horizontal representative error will likely affect the results less than other error sources, such as uncertainties in satellite retrieval, emission inventories, and model chemical mechanisms.
* * *
**Responses to the reviewer 3**

The review comments and our responses are coloured in blue and black texts, respectively. The changes in the manuscript corresponding to the comments are highlighted in yellow. The line and figure numbers refer to the revised manuscript.

In this study, Hoque et al. evaluated the global distribution of formaldehyde (HCHO) simulated by the CHASER v4.0 model against satellite, aircraft, and ground-based observations. Studies evaluating the global distribution of volatile organic compounds (VOCs) from models have been limited, and observations from space have the potential to help filling this gap. The investigations presented here add new insights, nevertheless, the manuscript in its current form has some limitations.

We thank the reviewer for the insightful comments, which has helped improving the manuscript quality.

The authors have already published a paper on the global distribution of HCHO from the CHASER model comparing with satellite data and MAX DOAS [Hoque et al., Atmos. Chem. Phys., 2022]. Sensitivity simulations analyzing the roles of different emissions have also been published there. In this case, a clear and detailed discussion is required (at the end of the introduction) on the main findings of that paper, the research gap, and the novelty of this new study.

**Response**: We have added the following texts in the revised manuscript.

L93-103 : Hoque et al. (2022) validated CHASER-simulated $NO_2$ and HCHO against OMI and MAX-DOAS observations for 2017. CHASER showed good skills in reproducing the OMI- (spatial correlation ® = 0.74) and MAX-DOAS- (temporal correlation R> 0.80) observed HCHO abundances. The study found that biomass burning contributes ~50% to the HCHO levels observed at the site in Thailand. However, the limitations of the study are: (1) Simulated HCHO partial column and profile were evaluated against MAX-DOAS observation on a seasonal scale only, (2) Model sensitivity studies were site-specific, thus providing no global statistics on emission contribution, and (3) Satellite observations were used as supporting datasets; thus the model-satellite comparison has not been comprehensive. This study utilizes multi-satellite (TROPOMI and OMI) HCHO observations, different $NO_x$ emission inventories, aircraft measurements, and daily and diurnal MAX-DOAS data to provide robust and comprehensive statistics on the model HCHO simulations.

Abstract: l.19: "CHASER reproduced the observed….", which observational data you are referring to?

**Response:** We have revised the sentence as follows:

CHASER reproduced the ==TROPOMI-observed== global HCHO spatial distribution with a spatial correlation (*r*) of 0.93 and a negative bias of 7%.

**Introduction**

"ozone production regime can be determined". In this context, you are referring to your past study. References where this type of approach was proposed [Martin et al., GRL, 2004] and later applied [Duncan et al., Atmos. Environ., 2010] should also be cited. Additionally, l.57-58: Several satellite-based observations have been used to evaluate the model simulation of HCHO by Chutia et al., [Environ. Poll., 2019]

**Response:** We thank the reviewer for the suggestion. We have included the references in the revised manuscript.

l.59-65: I agree that higher resolution TROPOMI may provide new features at finer resolution (3.5 km x 5.5 km). But how does it help your study running model at roughly 300 km x 300 km? Satellite data also seems to be averaged to the same grid resolution as the model, although that is not described in detail.

**Response:** We have added the following texts in response to this comment

==L196-205 TROPOMI observations are averaged spatially and temporally to the CHASER grid (T42) daily, leading to horizontal representativeness errors. However, the random horizontal representativeness errors are in the order of 5-10%, which is lower than the individual retrieval error of the satellite observations (Boersma et al., 2015). If the model horizontal resolution is increased by 50% (i.e., simulated at a horizontal resolution of 1.4° × 1.4°), the change in HCHO abundances is less than 6% (Fig S1 and Table S1 in supplementary information). The vertical sensitivity of the satellite retrievals is the most relevant source of representativeness error (Boersma et al., 2015). The current study utilizes the TROPOMI AK information to minimize the representativeness error. Therefore, the horizontal representative error will likely affect the results less than other error sources, such as uncertainties in satellite retrieval, emission inventories, and model chemical mechanisms.==

l.72-74: Mention what has been learned from these studies, possibly the quantitative role of anthropogenic emissions.

**Response:** We have revised the sentences as follows:

HCHO products have been used to infer changes in the global HCHO levels during the COVID-19 pandemic-led shutdown (Level et al., 2022; Souri et al., 2021; Su et al., 2021), ==demonstrating the role of anthropogenic emission on global HCHO variability.==

**Section 2: Model**

Anthropogenic emission is representative of which year. How has it been varied for different simulation years (2019, 2020)?

Response: we have revised the sentences as follow:

==L129-137 Anthropogenic $NO_x$ emissions for 2018 are obtained from the HTAP_v3 inventory (Crippa et al., 2023). Other anthropogenic emissions are taken from the HTAPv2.2 for 2008 and the biomass burning emissions from MACC-GFAS (Inness et al., 2013). The monthly soil $NO_x$ emissions derived from Yienger and Levy (1995) are constant each year. Biogenic emissions of VOCs are obtained from a process-based biogeochemical model: the Vegetation Integrative Simulator for trace gases (VISIT) (Ito and Inatomi, 2012). VISIT is a part of the CHASER modeling framework and incorporates the biogenic flux estimate scheme of Guenther et al. (1997) (Ito et al., 2022). The global isoprene emissions in VISIT and CAMS global biogenic emission inventory (Sinderolova et al., 2022; based on MEGANv2.1) are 400 and 450 TgC/yr, respectively.==

l.111-112- Seems ambiguous. The reanalysis data might not have provided an emission inventory. Maybe there is some inventory from the same or similar project.

**Response:** We have revised the sentences as provided in the earlier response.

l.113- The VISIT model is used here for estimating the flux of biogenic VOCs. How do these estimates compare to other widely applied MEGAN model (Guenther et al., Atmos. Chem. Phys., 2006) based inventories? ECMWF's CAMS has made freely available inventory for biogenic emissions, which may be used for comparison.

**Response:** We have revised this section and incorporated additional information of the biogenic flux.

VISIT is a part of the CHASER modeling framework and incorporates the biogenic flux estimate scheme of Guenther et al. (1997) (Ito et al., 2022). The global isoprene emissions in VISIT and CAMS global biogenic emission inventory (Sinderolova et al., 2022; based on MEGANv2.1) are 400 and 450 TgC/yr, respectively.

Table 1: Some simulations are missing from this list, like in which biogenic / biomass-burning is switched OFF.

**Response:** We have revised Table 1. And included all the simulations used in the study

l.196: Check and correct "TOGO" to "TOGA"

**Response:** We have corrected the typo error.

**Results**

Results from two different years appear identical in Figure 1. In the text also, there is no significant discussion on interannual differences. Values of correlation and RMSE have turned out to be the same between the analysis for 2 years separately. I suggest combining and discussing averages of both years for better statistics and reducing extra figures. This will further make this analysis consistent with follow-up results (e.g., Figures 2, 3), where mean is presented instead of year-wise segregation. The size of the figures can be enhanced.

**Response:** We thank the reviewer the comment. We also agree with the reviewer's perspective. We have revised (enlarged) Figure 1, but decided to retain the comparison for individual years. The reasons are-

(1) We believe, it is important to demonstrate the model-satellite agreement in both years to ensure consistency of our model.

(2) We have used one year of simulation in the sensitivity studies. Thus, information on the model performance in the individual years is important for supporting the results.

Figure 2 and other results: you are referring to different regions of the world. These need to be marked clearly on the global distribution map (Figure 1). Make bigger figures and define the regions on them.

**Response.** We thank the reviewer for this important suggestion. We have included a new figure (Fig.S3) in the supplementary information showing the region of interests. Moreover, we have redefined the regions of C-Africa, Europe, and India to be consistent with earlier studies with CHASER. We have also recalculated the statistics for these regions and made

appropriate corrections in the manuscript. There redefinition did 'not change the statistics and discussion significantly.

l.293 (and throughout the manuscript), be careful to always mention if you are referring to "spatial" or "temporal" correlations while reporting r values.

**Response:** We thank the reviewer for the important comment. We have ensured consistency in the correlation reporting throughout the manuscript.

l.300-301: This needs some supporting analysis/discussions. Either compare the emission inventory used here with other estimates or discuss if the model is underestimating particularly near urban centers (so to attribute to anthropogenic) but performing better in remote / vegetated areas.

**Response:** We have included the following sentences to address this comment

L335-338 NMVOC emissions from these sources (i.e., vehicular exhaust, solvent usage, and transport) are considered in the HTAPv2.2 inventory (Crippa et al., 2023). Although CHASER considered HCHO production from the degradation of anthropogenic VOCs, it is likely underestimated, resulting in a lower simulated winter-time HCHO column in this region.

Table 3 and other places: Are the temporal correlations derived from the mean seasonal cycle (12 points)? It is advisable to use all data (daily values over 2 years) to comment on temporal correlations. Or to discuss both ways. This is a "model evaluation paper" and these details are important.

**Response:** Yes, the temporal correlation has been calculated from the seasonal variation. We have included the temporal correlation estimated from the daily values in Table S2(supplementary information).

Table 4 and l.450: here also, clearly write if these are spatial correlations, seasonal (or daily). Check and make this aspect clear throughout the manuscript.

**Response:** We have included appropriate changes in the manuscript to address this comment.

l.471-472: here also check from MEGAN model-based emissions.

**Response:** The OMI-based top-down estimate is based on the MEGAN model. We have revised the sentence as follows:

L517-519 In CHASER, annual isoprene emissions over Amazonia are 67 Tg/yr, consistent with the OMI-based top-down estimates of 70 Tg/yr, estimated using apriori emissions from

Figure 5: I did not get the rationale behind enhancing anthropogenic emissions by a factor of 3. While HCHO was underestimated in reference simulation, now with this change the levels are equally (or more) overestimated over China, US, Africa, America (also see table 5). What has been achieved in terms of model performance?

**Response:** We thank the reviewer for the comment. We perturbed the anthropogenic VOC emissions to assess it's effect on the model-satellite comparison. Multiple simulation was performed and we found that, the perturbation effect is relevant when the anthropogenic VOC emissions are increased at least three-fold. Thus, we selected the lowest value (i.e., three-fold increase). If the perturbed simulation improves the model-satellite agreement, it can be interpreted as underestimated anthropogenic VOC emissions in standard simulation. Section 3.3 has been revised to address the reviewer comments.

l.503-504: No, the MBE values have increased! Check and revise/strengthen this whole section l.503-518. Also, reconsider tuning the simulation design itself (in place of 3 times more emissions)

**Response:** We have revised section 3.3 following the reviewer's earlier comment.

Section 3.4: This is an important aspect. Errors in the NOx emissions could have impacted model performance, especially in regions like South and Southeast Asia where inventories have greater uncertainties. Your simulations show that the model driven by older inventory shows lower bias in HCHO. Do you conclude that NOx emissions are overestimated in the new inventory? I did not find a clear assessment out of this important exercise.

**Response:** We have revised this section as follows:

L610-628The standard HCHO columns in India, China, and Southeast Asia are approximately 10–20% lower than the OLNE estimates (Fig.6(c)). In fact, those differences are consistent with changes in the regional OH estimates (Fig.6(d)). This finding implies that the changes in the $NO_x$ emissions estimates have affected the OH and HCHO abundances in these regions. Satellite data assimilation results reported by Miyazaki et al. (2017, 2020) indicate that $NO_x$ emissions in India have increased by 30% since 2008, whereas $NO_x$ emissions in China have declined since 2011 (Liu et al., 2016). Over E-China (Fig. 6(a &b)), the standard simulations reduce the absolute annual mean difference between OLNE and TROPOMI of $3 \times 10^{15}$ molecules cm$^{-2}$ to $1 \times 10^{15}$ molecules cm$^{-2}$, which is consistent with the lower $NO_x$ emissions in this region in the updated inventory (Fig . S8). Over India and SE-Asia, the standard OH concentrations are ~40% lower (Fig.6(d)) than the OLNE estimates, resulting in lower HCHO columns. The lower standard HCHO columns can be linked to the increasing $NO_x$ emissions in these regions (Fig.S8); however, the magnitude of

the change in the NO$_x$ emissions for these regions in the updated inventory is likely overestimated.

In E-USA and W-USA (Table S3), the standard simulation reduces the MBE by 26% and 12%, respectively. The reduction in MBE and RMSE values in Africa and South America is less than 10%. Therefore, NO$_x$ emission uncertainties mainly affect the HCHO simulations in India and SE Asia.

Tables and figures coming afterward often has data shown in previous figures and tables. Review them carefully and combine them whenever possible. Like, instead of comparing 1 simulation then another, you may put them in same table as reference, simulation1 and 2;

**Response:** We have removed the redundant tables

CHASER and TROPOMI are already compared (Section 3.1). Then there is an extra section comparing CHASER, TROPMI with OMI. Better to combine and strengthen the discussion.

**Response:** We have revised the subsection headings to avoid confusion.

Fig 8: When emission is increased (OLNE), why HCHO is reduced over Chiba and Kasuga, what is the underlying chemistry?

Response: This has been discussed in the manuscript as follows:

L812-817 Although the bias between OLNE and standard simulations for Chiba and Kasuga is ~4%, the absolute difference is ~1×10$^{15}$ molecules cm$^{-2}$. NO$_x$ emissions in Japan have not changed markedly since 2005 (Miyazaki et al., 2017). The differences between the simulations are observed during the summer when isoprene emissions are expected to peak (Hoque et al., 2018a). Because the OH estimates over Japan are similar for both simulations (Fig. 6(d)), the differences are likely related to the interaction between isoprene and NO$_x$ inventories.

Outside Japan also, there have been MAX-DOAS measurements. This paper being a global model evaluation, comparison over other regions of the world should also be added. If systematic data is not available, mean values may be compared (see Table 2 of Oomen et al., Atmos. Chem. Phys., 2024). Authors themselves have also published observations from another station in South Asia [Hoque et al., SOLA, 2018]

**Response:** We thank the reviewer for the comment. We are currently unable to include more A-SKY sites for the following reasons:

(1) Most of the A-SKY sites outside Japan were established after 2020, which is beyond the temporal limit of the current study. Moreover, the retrieved quantities for these sites are being investigated in detail in a separate project.

(2) Due to the mountainous terrain, we excluded the Pantnagar site (i.e., Hoque et al., SOLA, 2018).

However, we have included a comparison with the reported MAX-DOAS values by Oomen et al (2024) as shown in Figure 9.

**Minor comments**

Check the consistency of r values between l.352 and in Table 3.

**Response:** We have made appropriate changes in the manuscript

The data selection criteria for TROPOMI have not been discussed.

**Response:** We have added the following text in the revised manuscript

L194-195The filtering criteria of the TROPOMI datasets are as follows: quality assurance value (QA)>0.6, solar zenith angle <70º, cloud fraction < 0.3, AMF > 0.1, and surface reflectivity <0.2.

Line 387, line no. 467 - Figure no. is missing

**Response:** The figure has been included which is FigS6

In Figure S3, the correlation between TROPOMI and CHASER HCHO columns is marked as r=1 (blue text).

**Response:** We have revised the figure, which is FigS4 in the revised manuscript

Figure numbering should be corrected. There is no figure 4.

**Response:** We have ensured consistency in the Figure numbers and the corresponding texts.

Table 7 , last column name should have been 'r-value (CHASER vs. OMI)'

**Response:** We have revised Table 7

Line 781, slope values inconsistent with the slope in figure 9.

**Response:** We have ensured consistency in the Figure and texts.
* * *
**Response to reviewer 4**

The review comments and our responses are coloured in blue and black texts, respectively. The changes in the manuscript corresponding to the comments are highlighted in yellow. The line and figure numbers refer to the revised manuscript.

This paper presents the evaluation of HCHO columns from the CHASER model against the TROPOMI, OMI, ground-based MAX-DOAS observations, and the CHASER HCHO vertical profiles against the Atom-4 flight dataset. The authors compare the modelled regional HCHO columns with the TROPOMI and the OMI HCHO columns and analysed the model-observation differences comprehensively. The authors also compare the modelled HCHO columns with the MAX-DOAS columns at three locations in Thailand and Japan respectively. The modelled HCHO profile and the profile from the Atom-4 flights are compared for Amazonia and for the Remote Pacific region, respectively. The authors have also performed sensitivity simulations to assess the impact of anthropogenic, biogenic, and biomass burning VOC emissions, as well as NOx emissions on modelled HCHO. However, I find that one limitation is the lack of discussions on the important role of chemical mechanisms in simulating HCHO in the models, despite that the authors did mention this in the conclusion. There are some previous studies that the authors could cite which addressed inter-model differences in modelled HCHO (see below suggestions) Overall, the analysis is thorough and robust. The paper is generally well-written, and the materials are well organised, and is within the scope of GMD. However, the presentation of the paper can be improved. I encourage the authors to make a thoroughly revision of the manuscript.
Below are two relevant papers on model differences in modelling HCHO (and CO):

Anderson, D. C., Nicely, J.M., Wolfe, G. M., Hanisco, T. F., Salawitch, R. J., Canty, T. P., … Zeng, G. (2017). Formaldehyde in the tropical western Pacific: Chemical sources and sinks, convective transport, and representation in CAM-Chem and the CCMI models. Journal of Geophysical Research: Atmospheres, 122. https://doi.org/10.1002/2016JD026121 (Figure 13)
Zeng, G., Williams, J. E., Fisher, J. A., Emmons, L. K., Jones, N. B., Morgenstern, O., Robinson, J., Smale, D., Paton-Walsh, C., and Griffith, D. W. T.: Multi-model simulation of CO and HCHO in the Southern Hemisphere: comparison with observations and impact of biogenic emissions, Atmos. Chem. Phys., 15, 7217–7245, https://doi.org/10.5194/acp-15-7217-2015, 2015. (Figure 15 and Table 4)

**Response:** We thank the reviewer for the insightful comments which has helped improving the quality of the manuscript. We have included additional discussion on the role of chemical mechanism on the simulated HCHO.

**3.9 Uncertainties in the chemical mechanism**

Uncertainties in the chemical mechanisms affect the HCHO simulations. Representation of isoprene chemistry can vary among the gas-phase chemistry mechanisms used in the CTMS. The most commonly used isoprene schemes underestimates observed HCHO by at least 15% (Marvin et al., 2017). Such underestimations are also strongly linked with the errors in the $NO_x$ emission inventories (Anderson et al., 2017). In addition, potential errors in the acetaldehyde emission and chemistry can also lead to underestimated HCHO vmr up to 75 pptv in the lower troposphere (Anderson et al., 2017).

**My specific comments are listed below.**

Abstract
It feels that the abstract is overly concise and does not reflect fully what are presented in the paper.

**Response:** We have revised the abstract

L19-20: Please state which comparison this is for, i.e., TROPOMI.

**Response :** We have revised accordingly

L30: It is the comparison between the CHASER and MAX-DOAS HCHO columns, not mixing ratio. Please also state the disagreement, i.e., CHASER underestimates the HCHO peak in comparison with the MAX-DOAS data at all three locations. You speculate that the model data averaged over a large area might not be able to capture the observed peak at these locations. A mention of this would be useful in the abstract.

**Response:** We have revised the whole abstract following the reviewer's earlier comment. Such information is included in the revised abstract.

Introduction
L82: How do you evaluate OH?
**Response:** The OH was validated against Atom observation by Sekiya et al., (2018)

Model, observations, and methods
L93: Is there a reference for this?
**Response;** Reference has been included

L99-101: A list of the reactions in a table (in supplementary) could be considered if they have not been published before.

**Response:** The reactions has been published by Sudo et al., 2002 (Reference provided in the reference section of the manuscript)

L110-123: It will be helpful to tabulate these emissions.
**Response**: We agree with the reviewer's perspective. However, to reduce the number of tables, we decided to describe the emission inventories.

L117: Do you calculate lightning NOx emissions online or prescribe them?

**Response:** The lightning NOx emissions were calculated online using the widely-used cloud-top height (CTH) scheme (Price and Rind, 1992).

L124-125: Are there OH observations from OMI and Atom? Please provide details.

**Response:** OH observations are only available from the Atom campaign, We have revised the sentences as follows:

L146-147 Sekiya et al. (2018) comprehensively assessed CHASER simulated $NO_2$ abundances using OMI observations. CHASER well reproduced the ATom-observed OH spatiotemporal variation (Sekiya et al., 2018).

Table 1: ANI and OLNE appear first time in Table 1. Please define these simulations in the text.
**Response:** Table 1 has been revised. All the abbreviations are defined.

L137: What are the TROPOMI grids?
**Response:** We have removed the sentence

L139: Do you mean that the TROPOMI data are interpolated onto the CHASER horizontal grid?
**Response:** TROPOMI data has been interpolated onto the CHASER horizontal grid. We have included additional discussion in the revised manuscript as follows:

L196-205 TROPOMI observations are averaged spatially and temporally to the CHASER grid (T42) daily, leading to horizontal representativeness errors. However, the random horizontal representativeness errors are in the order of 5-10%, which is lower than the individual retrieval error of the satellite observations (Boersma et al., 2015). If the model horizontal resolution is increased by 50% (i.e., simulated at a horizontal resolution of 1.4° × 1.4°), the change in HCHO abundances is less than 6% (Fig S1 and Table S1 in supplementary information). The vertical sensitivity of the satellite retrievals is the most relevant source of representativeness error (Boersma et al., 2015). The current study utilizes the TROPOMI AK information to minimize the representativeness error. Therefore, the horizontal representative error will likely affect the results less than other error sources, such as uncertainties in satellite retrieval, emission inventories, and model chemical mechanisms.

**Response :** We have revised accordingly

Results and discussion
L231: This section is essentially the comparison of CHASER HCHO with TROPOMI. Maybe "TROPOMI" should be reflected in the section title?

**Response:** We have revised the section title to : Comparison of CHASER HCHO with TROPOMI observations

L235-239: I am not sure how meaningful these statistics are in terms of the global means as the global HCHO distribution is so inhomogeneous.

**Response:** Similar statistics were reported in earlier studies on CHASER simulations (Hoque et al., 2022; Sekiya et al., 2018; Ha et al., 2022, He et al., 2023)

L243-245: Would it more suitable to note this in the MAX-DOAS comparison section?

**Response:** We agree with the reviewer's perspective. However, we think it is also important to mention here to support the uncertainties in the observations and simulations.

Table 2: These numbers don't have to be in a table. You could include them in the Figure 1 caption. Is the correlation coefficient spatial or temporal?

**Response:** We agree with the reviewer's perspective. However, we think the table is important to provide vital information on the global statistics for the readers, without checking the details. We also believe it is important for citation of our work too.
These are spatial correlation and has been described in the caption of table 2.

Figure2: The panels can be larger. Mark the position of the MBE numbers in the panels consistently. Add identifiers to the sub-figure, e.g., (a), (b), … for each region. Then refer to Figure2(a), Figure 2(b), etc., when you discuss them in the following subsections.

**Response:** Figure 2 has been revised accordingly. We have also ensured the consistency of the figure numbers in the revised manuscript.

L293-: Please refer to the figure(s) and table(s) that your discussions are based on at the beginning of each subsection. Same as the following subsections of (b), (c), etc.

**Response:** Following the reviewer's comment we have ensured mentioning the figure numbers in he text

L300: Do you mean direct HCHO emissions or indirect (degradation of VOCs) HCHO emissions? Please clarify.

**Response:** We have added the following text to address this comment

L335-338 NMVOC emissions from these sources (i.e., vehicular exhaust, solvent usage, and transport) are considered in the HTAPv2.2 inventory (Crippa et al., 2023). Although CHASER considered HCHO production from the degradation of anthropogenic VOCs, it is likely underestimated, resulting in a lower simulated winter-time HCHO column in this region.

L319: Can you speculate what drives these model-satellite discrepancies in the Europe and W-US in summer and autumn?

**Response:** We have added the following text in response to this comment

L358-365 In both regions (i.e., Europe and W-USA), the biogenic and anthropogenic contribution to the total HCHO level is equivalent during summer. In autumn, the anthropogenic emission contributions are higher. (Section 3.8). This manifests a potential model underestimation of biogenic HCHO levels in these regions, linked to the uncertainties in the biogenic emission inventory and isoprene mechanism

L328-329: Please refer to the figure you are referring to. Please also note that the C-Africa off-peak HCHO is overestimated by CHASER compared to TROPOMI (Figures 1 and 2).

**Response:** We have ensured consistency in the figure numbers throughout the manuscript. We have also revised the discussion on C-Africa.

L333: Figure S4: There is no black curve in Figure S4. Please revise this figure or the caption.

Response: Figure S4 has been revised which is FigS5 in the revised manuscript

L338: Please mention the figure you refer to for these discussions.

**Response:** The figure number has been included.

L339: Missing ")" in "(De Smedt et al., 2008". Again, please mention the figure here you are referring to over the next few lines.

**Response:** The figure number has been included and the consistency has been ensured throughout the revised manuscript.

L340-341: "The lower CHASER columns in Southern Africa are likely attributable to underestimated pyrogenic emissions." - Can you confirm this from the following sensitivity simulations?

**Response:** We have revised the sentence as follows:

L382-385Over South Africa (S-Africa; Fig.2(g)), elevated TROPOMI HCHO columns are consistent with GOME-2 and SCIAMACHY observations (De Smedt et al., 2008). The observed peaks in HCHO columns and FRP values (Fig.S5) are consistent and thus can be attributed to biomass burning. Pyrogenic emissions contribute ~36% to the high HCHO columns in this region (section 3.8).

L355-356: Why particularly mention the biomass burning in N Africa here?

**Response:** N-Africa is mentioned to compare between two biomass-prone regions. We have revised the sentence.

L358-359: Could the chemical mechanism in the model be at play?

**Response:** We have revised the sentence as follows:

L41-403 The model overestimates the HCHO columns in S-America, similarly to C-Africa and N-Africa, probably because of the uncertainties in biogenic emission inventories and the isoprene oxidation scheme.

L370: Refer to relevant figure(s) and table(s) earlier in your discussion of the results. Do you have an estimate how the biogenic and biomass burning emissions in India compared to other regions? Are there any specific meteorological conditions in this region that lead to low HCHO and the lack of seasonality?

**Response**: We have revised this section as follows to address the comment.

L406-419 CHASER well reproduced the observed HCHO spatial distribution in India ( Fig.2 (i); $r$ =0.84), with MBE and RMSE of $-1.20 \times 10^{15}$ and $1.775 \times 10^{15}$ molecules cm$^{-2}$. However, the temporal correlation ($R$=0.18) between the datasets is low. The observed seasonal modulation of ~30% manifests a less-prominent seasonality in HCHO abundances in India. The correlation between temperature variations and isoprene emissions in India is inhomogeneous (Starvakou et al., 2014). India has a diverse landscape, including major forests over the east, northeast, and southwest regions and deserts in northwestern India (Surl et al., 2018). The Indo-Gangetic Plain (IGP) stretches from Eastern Pakistan to Bangladesh and is a major agricultural region in India (Kuttippurath et al., 2022). Thus, averaging the HCHO columns over a diverse landscape can lead to a less prominent seasonality. Moreover, biomass burning compromises 23% of India's total NMVOC (13 Tg/yr) emissions (Stewart et al., 2021). Sensitivity analysis (section 3.8) estimates show biomass burning contribution

to the HCHO levels in India is ~2%., manifesting that the modeled biomass burning emissions for India are underestimated. Considering the diverse Indian landscape, the model satellite comparison over three regions in India (IGP, east India, and South India) is shown in Fig.2 (j-l).

L387: The figure number is missing here.

**Response :** We have included the figure number.

L422-423: I am not sure what you try to convey here?

**Response:** The sentence has been removed for clarity.

Figure 5: Could you increase the size of the panels in this figure?

**Response:** We have revised all the figures

L552: It is important to summarise the NOx emissions in the two inventories you used. What are the differences in NOx emissions between these two inventories? It will help to understand the impact of NOx on HCHO and OH.

**Response:** The differences are mentioned in text. The NOx emissions are shown in Fig.S8. The discussion has been revised.
L605 – 628 The differences between the two $NO_x$ inventories are – (1) HTAP-v3 inventory considers the changes in $NO_x$ emissions from 2000 to 2018, whereas the temporal coverage of HTAP_v2.2 is 2008 – 2010, and (2) Emissions in HTAP-v3 have a higher sectoral disaggregation (Crippa et al., 2023). The comparison-related statistics are given in Table S3. $NO_x$ emissions from both inventories are shown in Fig. S8

On a global scale, HCHO column estimates are mostly unaffected by the changes in the $NO_x$ emission inventories, manifested by the MBE values (Table 6). However, RMSE is 8% lower in the case of standard simulation. OLNE estimates in the higher latitude (>=50ºN) are 5% lower than the standard simulations. Such differences do not affect the model–satellite agreement in these regions.

The standard HCHO columns in India, China, and Southeast Asia are approximately 10–20% lower than the OLNE estimates (Fig.5(c)). In fact, those differences are consistent with changes in the regional OH estimates (Fig.6(d)). This finding implies that the changes in the $NO_x$ emissions estimates have affected the OH and HCHO abundances in these regions. Satellite data assimilation results reported by Miyazaki et al. (2017, 2020) indicate that $NO_x$ emissions in India have increased by 30% since 2008, whereas $NO_x$ emissions in China have declined since 2011 (Liu et al., 2016). Over E-China (Fig. 5(a &b)), the standard simulations

reduce the absolute annual mean difference between OLNE and TROPOMI of $3 \times 10^{15}$ molecules cm$^{-2}$ to $1 \times 10^{15}$ molecules cm$^{-2}$, which is consistent with the lower NO$_x$ emissions in this region in the updated inventory (Fig . S8). Over India and SE-Asia, the standard OH concentrations are ~40% lower (Fig.5(d)) than the OLNE estimates, resulting in lower HCHO columns. The lower standard HCHO columns can be linked to the increasing NO$_x$ emissions in these regions (Fig.S8); however, the magnitude of the change in the NO$_x$ emissions for these regions in the updated inventory is likely overestimated.

In E-USA and W-USA (Table S3), the standard simulation reduces the MBE by 26% and 12%, respectively. The reduction in MBE and RMSE values in Africa and South America is less than 10%. Therefore, NO$_x$ emission uncertainties mainly affect the HCHO simulations in India and SE Asia.

L553: you need to define the OLNE simulations before referring to it.

**Response:** The abbreviations of the simulations have been defined in Table 1 in the revised manuscript.

L565-566: Which figure that you are referring to here?

**Response :** We have included the figure number.

Figure 6: It will be helpful to understand this figure if the di□erences in NOx emissions are displayed or mentioned.

**Response :** The differences are mentioned in text. The NOx emissions are shown in Fig.S8. The discussion has been revised.

L595-: This section should be condensed where appropriate. You have compared CHASER and TROPOMI HCHO columns in detail already, so the focus here should be on what those most significant di□erences between OMI and TROPOMI HCHO are and how they compare with the CHASER HCHO.

**Response:** We removed the redundant information.

L612: Referring to Figure 7 at the beginning of this paragraph.

**Response:** We have revised the figure numbers in the revised manuscript and ensured consistency throughout the manuscript

L680: Which "observation" do you refer here?

**Response:** Both OMI and TROPOMI. We have revised the sentence.

L688-689, L703: what are differences between OMI SOA and OMI BIRA HCHO products? A brief introduction will be helpful.

**Response:** We have added the following text to address this comment

L728-731 The most relevant differences between the OMI BIRA and SAO products are related to the underlying CTMs that simulate the apriori profiles and the reference sector correction (Zhu et al., 2016). A comprehensive list of the differences between the two products is available from Zhu et al. (2016).

**Response:** Yes, coincident data were used for the comparison

L808: "In Kasuga, modelled diurnal variations correlate strongly (R=0.85) with the observations". But in Figure 9, the R value is 0.41, not 0.85. Please check.

**Response:** We have revised the texts.

L837-839: Could you elaborate a bit more on this mechanism?

**Response:** We have added the following texts.

L904-909 Zhao et al. (2022) reported a similar finding and attributed enhanced $CH_4$ oxidation in the presence of water vapor to the HCHO mixing ratios above 2 km. At higher altitudes HCHO is produced through the $CH_4$ oxidation (i.e., $CH_4 + OH$) initiated $CH_3O_2$ (methyl peroxy radical) + $CH_3O_2$ pathway. HCHO production through this pathway is considered in CHASER Therefore, despite the differences in the magnitude, CHASER has shown good skills in reproducing the VOC profiles.

**Response:** We have revised the texts. We agree with the reviewer's perspective of providing the box. However, we adopted the style of earlier studies (i.e., He et al., 2022, Sekiya et al., 2018) using ATom measurements.

L872-873: Have you already defined these sensitivity simulations?

**Response:** We have defined all the simulations in the Table 1 in the revised manuscript

L932-934: Does the model's course resolution play a role in this case?

**Response:** We didn't infer the impact of model resolution on this comparison. However, our earlier studies (i.e., Hoque et al., 2022) have demonstrated the impact of horizontal resolution on the model-ground-base comparison.

L944: The last half sentence doesn't read well; do you mean the model underestimates the biogenic contributions?

**Response:** We have revised the sentence

L1020-1023 Lastly, sensitivity studies were conducted to estimate the contributions of the different emissions sources to the total HCHO columns in different regions. Biogenic emissions were the most significant contributor in most of the regions. In a few cases, biogenic and anthropogenic emission contributions were equivalent. In some regions, only summertime biogenic estimates were found to be reasonable.